



# The estimates of carbon sequestration potential in an expanding Arctic fjord affected by dark plumes of glacial meltwater (Hornsund, Svalbard)

Marlena Szeligowska[1], Déborah Benkort[2], Anna Przyborska[3], Mateusz Moskalik[4], Bernabé Moreno[1], Emilia Trudnowska[1], Katarzyna Błachowiak-Samołyk[1]

[1]Marine Ecology Department, Institute of Oceanology, Polish Academy of Sciences, Sopot, 81-712, Poland
[2]Institute of Coastal Systems - Analysis and Modeling, Helmholtz-Zentrum Hereon, Geesthacht, 21502, Germany
[3]Physical Oceanography Department, Institute of Oceanology, Polish Academy of Sciences, Sopot, 81-712, Poland
[4]Department of Polar and Marine Research, Institute of Geophysics, Polish Academy of Sciences, Warsaw, 01-452, Poland

*Correspondence to*: Marlena Szeligowska (lena@iopan.pl)

**Abstract.** In polar regions, glaciers are retreating onto land, gradually widening ice-free coastal waters which are known to act as new sinks of atmospheric carbon. However, the increasing delivery of inorganic suspended particulate matter (iSPM) with meltwater might significantly impact their capacity to contribute to carbon sequestration. Here, we present an analysis of satellite, meteorological, and SPM data as well as results of the coupled physical-biogeochemical model (1D GOTM-ECOSMO-E2E-Polar) with the newly implemented iSPM group, to show its impact on the ecosystem dynamics in the warming polar fjord (Hornsund, European Arctic). Our results indicate that with a longer melt season (9 days per decade, 1979-2022), loss of sea ice cover (44 days per decade, 1982-2021) and formation of new marine habitat after the retreat of marine-terminating glaciers (around 100 km$^2$ in 1976-2022, 38% increase in the total area), glacial meltwater has transported increasing loads of iSPM from land (3.7 g·m$^{-3}$ per decade, reconstructed for 1979-2022). The simulated light limitation induced by iSPM input delayed and decreased phytoplankton, zooplankton, and macrobenthos peak occurrence. The newly ice-free areas still markedly contributed to the plankton primary and secondary production, and carbon burial in sediments (5.1, 2.0, and 0.9 GgC per year, respectively, average for 2005-2009 in the iSPM scenario). However, these values would have been higher by 5.0, 2.1 and 0.1 GgC per year, respectively, without iSPM input. Since carbon burial was the least affected by iSPM (around 16% decrease in comparison to 50% for plankton primary and secondary production), the impact of marine ice loss and enhanced land-ocean connectivity should be investigated further in the context of carbon fluxes in expanding polar fjords.

## 1    1 Introduction

Organic carbon burial in marine sediments represents the dominant natural pathway toward long-term sequestration and hence plays a key role in controlling atmospheric $O_2$ and $CO_2$ concentrations (Berner, 1982; Hedges and Keil, 1995). While important carbon sinks at coastal wetlands (mangroves forests, salt marshes, and seagrass beds) are declining globally



(Duarte et al., 2005; Howard et al., 2014), new marine habitats are opening up in the Arctic and West Antarctic due to glaciers retreat and giant iceberg calving (Ficetola et al., 2021). Within these coastal ecosystems, enhanced underwater light conditions and nutrient supply from land increase $CO_2$ drawdown by phytoplankton and ice algae, hence intensifying the cascade from carbon capture into storage and burial in sediments (Ardyna and Arrigo, 2020; Arrigo et al., 2008; Wadham et al., 2019). Due to the high sedimentation rates, emerging and expanding fjords play an important role as efficient carbon burial hot spots (Bianchi et al., 2020; Cui et al., 2022; Smith et al., 2015). Thus, the loss of marine ice in polar coastal waters might to some extent compensate for decreasing coastal carbon sinks elsewhere (Barnes, 2017; Peck et al., 2010; Zwerschke et al., 2022).

Despite recent increases in primary and secondary production due to, among others, the earlier break-up of seasonal sea ice, the polar regions' potential for long-term carbon burial in sediments is ultimately limited by multifarious mechanisms. The changes in the duration and composition of ice algae blooms weaken the sympagic-benthic coupling, in consequence leaving more biomass that can be utilized and dispersed in the pelagic system (Fadeev et al., 2021; Lalande et al., 2019; Riser et al., 2008). Thus, warming induces the maturation of polar fjords i.e. the transition to a more complex but effective pelagic food web consuming most of the available organic matter, and thus less carbon is deposited at the bottom (Węsławski et al., 2017; Zaborska et al., 2018). Furthermore, the delivery of inorganic suspended particulate matter (iSPM) with glacial meltwater dims underwater light later in the productive season (summer and autumn) (Szeligowska et al., 2022) and results in a significant reduction of phytoplankton and phytobenthic biomass (Blain et al., 2021; Deregibus et al., 2016; Holt et al., 2016) that influences carbon burial potential in glacial bays.

Further warming will likely exacerbate sediment inputs through increasing precipitation, storm activity enhancing erosion, glacial melt, permafrost thaw, and sea-level rise (Syvitski et al., 2005, 2022). Moreover, in situ observations and numerical simulations from Arctic fjords suggest that after marine-terminating glaciers retreat onto land, subglacial discharge and nutrients upwelling ceases, enhancing surface stratification, and weakening vertical mixing, therefore reducing the productivity in coastal zones (Hopwood et al., 2018; Meire et al., 2017). However, our understanding of the rapid transformations of polar marine ecosystems under climatic stressors remains insufficient due to, among others, the scarcity of long-term standardized monitoring data (Schofield et al., 2010). While numerical models were essential in filling the knowledge gaps related to the mechanisms of nutrient supply with meltwater (Castelao et al., 2019; Oliver et al., 2020) and substantial effort has been put into incorporating modules representing biogeochemistry in sea ice (Steiner et al., 2016), only a few of them resolve inorganic particulate matter dynamics in glacier-fed basins (Neder et al., 2022). So far, these models do not typically represent the impact of the delivery of terrigenous material on biological production and carbon budgets.

This study aimed to assess the gains and losses in plankton primary and secondary production, and carbon burial due to the transformations of the European Arctic coastal waters. We investigated Hornsund (Svalbard, West Spitsbergen) as a model high-latitude fjord, since it is among the most studied fjords in the Arctic and represents an area of rapid regional warming with many bays affected by the recession of glaciers. Thus, here we (1) map the extent of emerging habitat after the retreat of marine-terminating glaciers and (2) simulate how the ecosystem dynamics and carbon sequestration are affected by sediment



discharge in these bays using a 1D coupled physical-biogeochemical model (GOTM-ECOSMO-E2E-Polar) with newly implemented iSPM group. We present the results of our simulations for 2005-2009, i.e., a period with an exceptionally strong warming signal (Muckenhuber et al., 2016; Promińska et al., 2017), in the context of multidecadal (1976-2022) changes in the physical environment to discuss the potential of newly ice-free areas to act as emerging carbon sinks and their role in the global carbon cycle.

## 2     Methods

### 2.1     Study area

Hornsund is a glaciomarine fjord of Svalbard with inner basins affected by glacial outflow (Fig. 1) (Błaszczyk et al., 2019). Since the strong polar front formed by the West Spitsbergen Current (saline and warm Atlantic Water) and the Sørkapp Current (cold and relatively fresh Arctic Water) reduces the advection of Atlantic Water into Hornsund in comparison to

other West Spitsbergen fjords (Promińska et al., 2017), it is considered a less mature, highly productive cold-water fjord with an Arctic-type resident biota and relatively high sequestration of organic carbon (Węsławski et al., 2017; Zaborska et al., 2018). Characterized by dynamic paraglacial coastal systems with high sediment mobility, Brepollen is the most extensive bay in Hornsund, where >85 km of new shoreline was formed in the last century after an ice retreat (Strzelecki et al., 2020). The area is known for one of the fastest retreat rates of marine-terminating glaciers in the Svalbard archipelago, which has

accelerated in this century up to around 3 km$^2$ per year in 2001-2010 (Błaszczyk et al., 2013). Importantly, the ice bridge between Brepollen and Hambergbukta (Fig. 1b, currently <5km wide) is predicted to break up in the coming decades (Grabiec et al., 2018; Osika et al., 2022), thus reopening a direct connection to the Barents Sea and changing the hydrodynamic conditions for biological production and carbon burial by either stronger sea ice or Atlantic Water advection.

### 2.2     Datasets

#### 2.2.1     The area and volume of newly ice-free marine habitats

Summertime Landsat images of Hornsund were downloaded from https://glovis.usgs.gov/app (Sup. Tab. 1). Only cloud-free images with no sea ice cover (from July to early September) were used. When present, 4-3-2 and 3-2-1 spectral bands (Landsat 8 and Landsat 1-7, respectively) were used to prepare RGB composites, and a panchromatic band (8) was used to enhance the resolution. Newly ice-free areas were manually delineated with the position of glacier fronts in 1976 as

a reference since it was the first year with Landsat images available in summer (Fig. 1b). The same person (MS) repeated the procedure three times for each year to test the repeatability of manual digitization. The standard deviation was up to 0.23 km$^2$. Importantly, the fronts of marine-terminating glaciers undergo seasonal fluctuations, which might increase uncertainty (Błaszczyk et al., 2021, 2023). However, here we narrowed the analysis to the main melt season (from July to early September). Marine habitat volume was calculated based on digitized area and bathymetry data from Hornsund (grid size



100 m) (Moskalik et al., 2014) (data available until glacial fronts extent in 2010) using the zonal statistics method in ArcGIS Pro 2.8.0.

### 2.2.2    Sea/ice surface temperature and sea ice concentration

Arctic Sea and Ice Surface Temperature dataset delivered sea and ice surface temperature (SST) and sea-ice concentration (SIC) (L4, 5km, daily). These data were provided by Danish Meteorological Institute and MyOcean regional data assembly
centre and created using multisensor satellite surface temperature observations. Since the dataset did not cover all the fjord, data were extracted from points in the outer/central parts (Fig. 1, Sup. Tab. 2) assuming that they reflect the SST/SIC conditions in the inner fjord. This assumption is supported by previous studies (Arntsen et al., 2019; Błaszczyk et al., 2021; Sutherland et al., 2013) and the fact that this analysis focuses on relative changes in melt season intensity rather than absolute values. The data were extracted for three adjacent cells (Sup. Tab. 2) and averaged. Sea ice-free days (SIF) were
defined as a fraction of the year with SIC<15%. The monthly mean extent of SIC>15% in March 2005 and 2006 is shown in Fig. 1a. The sum of all daily SST>0°C (positive degree days, PDD SST) was calculated for each year (annual) and each melt season (summertime, June-August) as a proxy for submarine melt potential (Hock, 2005; Rignot et al., 2008).

### 2.2.3    Air temperature and precipitation

Air temperature and precipitation datasets from Polish Polar Station Hornsund (PPS, Fig. 1c) were downloaded from
Wawrzyniak and Osuch (2020) (1979-2018, https://doi.pangaea.de/10.1594/PANGAEA.909042) and from SIOS (https://doi.org/10.5194/essd-12-805-2020, 2018-2022). Daily average air temperature (AT) was used to calculate the sum of all daily AT>0°C (PDD AT) for each year (annual) and each melt season (summertime, June-August) as a proxy for surface melt potential (Hock, 2005; Rignot et al., 2008). Annual and summertime (June-August) precipitation was calculated by summing up the daily precipitation measurements (mm). The start of the melt season was defined as the start of the first
period of six consecutive days with AT>0°C; similarly, the end of the melt season was defined as the first of six consecutive days with AT<0°C (modified from Błaszczyk et al., 2021). It takes into account the delays in meltwater and particulate matter delivery to the fjord and the 6-days window was shown to be well correlated with sediment flux (this study and D'Angelo et al., 2018). Melt season duration was calculated as the number of days between the end and the start of the melt season and provided as a fraction of the year.

### 2.2.4    Suspended particulate matter, sediment flux, and salinity

Datasets for suspended particulate matter (SPM), sediment flux, and salinity collected in Hansbukta (2015-2021) at long-term monitoring stations (Fig. 1c, Sup. Tab. 2) were downloaded from https://dataportal.igf.edu.pl/group/longhorn. In this study, sediment flux data from sediment traps deployed for one day at 5, 10, 15, and 20 m depths were considered for analysis. Inorganic and organic SPM concentration (g·m$^{-3}$) and sediment flux (g·m$^{-2}$·day$^{-1}$) were calculated based on total
SPM/sediment flux and the loss on ignition (Moskalik et al., 2018). Integrated iSPM concentration in the water column was





averaged for the main melt season (June-August, 2016-2021). The annual variability in the SPM levels was visualized using a kernel density estimate (KDE) plot prepared based on SPM data (2015–2021) from discrete depths. The sinking rate of SPM (m·day$^{-1}$) was calculated by dividing sediment flux by SPM concentration sampled at corresponding depth layers (Mugford and Dowdeswell, 2011). Sediment flux and salinity datasets were used for model parametrization (see 2.3.1), whereas the SPM dataset was used for model assessment (see 2.4).

## 2.3    Numerical model

To study the dynamics of the West Spitsbergen coastal waters affected by iSPM input, numerical experiments were designed using the Polar version of the biogeochemical ECOSystem Model (ECOSMO-E2E-Polar version) coupled with the General Ocean Turbulence Model (GOTM) (Burchard et al., 1999). ECOSMO-E2E-Polar version represents the three main nutrient cycles (nitrogen, phosphorus, and silica) in the pelagic and sympagic systems, three functional groups of primary producers (ice algae, diatoms, and flagellates), two zooplankton groups (micro- and meso-), and one macrobenthos group and it was fully described by Benkort et al. (2020), Daewel et al. (2018), Daewel and Schrum (2013), Yumruktepe et al. (2022). We extended the biogeochemical model to include iSPM in the model formulation (Fig. 2). The model was built with the Fortran-based Framework for Aquatic Biogeochemical Models (FABM) (Bruggeman and Bolding, 2014) to facilitate coupling with the physical model. In this first application, a 1D numerical framework was used, and physical processes in the water column were calculated by GOTM. Simulation resolved the profiles of velocities, temperature, salinity, turbulent mixing, and transport of ecosystem state variables in 20 vertical layers with surface zooming of 1.5 and bottom zooming of 0.1. This approach neglects horizontal transport and considers vertical exchange processes only. It allows feasible parameterization, verification, and sensitivity tests to study processes with a low computational effort but hinders the model's skills to represent advection and upwelling. However, advection is considered to be limited in Hornsund in comparison to other West Spitsbergen fjords, in particular in the inner bays, whereas upwelling is most important near the glacier fronts (up to 500 m, Pasculli et al., 2020).

The model was implemented at 20 stations located within the newly ice-free areas in Hornsund (Fig. 1b, Tab. 1). The simulations were run from the beginning of 2005 to the end of 2009. Input data from 2005-2009 were averaged, repeated five times, and used as a spin-up to allow the model to reach equilibrium under the applied forcing. Temperature and salinity vertical forcing were used from the 3D hydrodynamic numerical model of Hornsund which represents 9 sources of freshwater input from the Hornsund drainage basin including all components (ablation, precipitation, snow, and rivers) (HMR, Jakacki et al., 2017). Sea-ice thickness and concentration were extracted from the S800 model simulation at the closest grid cells (Albretsen et al., 2017), the same data as used for the HRM model. Sea-ice input was smoothed using a 30-day rolling average, as the 1D setup does not represent advection, and highly variable thickness and concentration affect the performance of the ice algae module. The atmospheric conditions were prescribed from meteorological monitoring in the Polish Polar Station Hornsund, i.e. air temperature (2m above the surface), eastward (u) and northward (v) wind speed, cloudiness, relative humidity, and pressure. The model was run with a 30-minute time step and the daily average was saved





as an output. Two sets of scenarios were performed to evaluate the gains in carbon sequestration potential due to the retreat
of marine-terminating glaciers, and losses due to the iSPM discharge by evaluating plankton primary and secondary
production and carbon burial. The SPM scenario included the iSPM input prescribed to the model according to Eq. (1) and
noSPM scenario was a control run without iSPM input.

### 2.3.1 ECOSMO developments

Two state variables were added to the ECOSMO-E2E-Polar model framework, accounting for iSPM and iSPM sediment
pool (sed$_{iSPM}$) (Fig. 2). The input of iSPM prescribed to the model was calculated based on the inorganic sediment flux (iSF)
and its relationship with air temperature and salinity (Sup. Fig. 1) developed from field data in a form of Eq. (1):

$$iSF = 10^{0.04 \cdot 6accPDD\ AT + 0.174 \cdot (refS - meanS) + 0.815} ,$$ (1)

where 6accPDD AT is the accumulated daily air temperature for positive degree days for a 6-day window (°C), refS is a
reference salinity for Atlantic Water (34.9) (Moskalik et al., 2018), and meanS is the mean salinity above the sediment trap.
The ordinary least squares (OLS) function (statsmodels library in Python) was used to generate a linear model for iSF
estimates. The root-mean-square deviation (RMSD) was used to measure the differences between the observed iSF and
linear model. iSF calculated for each depth layer was prescribed to the model as a daily input ($C_{iSPMinpput}$) in mg·m$^{-3}$. While
some runoff data are available for Hornsund (Van Pelt et al., 2019; Błaszczyk et al., 2019), here it was not feasible to
parametrize the iSPM input based on the meltwater discharge due to the structure of the hydrodynamic model (1D in contrast
to 3D) and lack of data on sediment loads in glacial plumes. However, in this study, salinity depended on the discharge
provided in the 3D hydrodynamic model that was a source of input data (HMR, Jakacki et al., 2017). Therefore, we used the
salinity as a proxy of the inorganic sediment.

The state variables in ECOSMO (list of all the state variables in Sup. Tab. 3) are solved using prognostic equations in the
form of Eq. (2):

$$C_t + (w_d)C_z = (A_v C_z)_z + R_C,$$ (2)

with $C_x = \frac{dC}{dx}$, where x represents either time (t) or depth (z). The equation includes vertical turbulent subscale diffusion,
sinking rates and chemical and biological interactions. The vertical turbulent sub-scale diffusion coefficient ($A_v$) is estimated
by the hydrodynamic core of ECOSMO. The sinking rate ($w_d$) is a constant, non-zero only for detritus, opal, and iSPM. The
sinking rate ($w_d$) applied in the model that allowed to properly represent the dynamics of iSPM was 0.8 m·day$^{-1}$, which is a
lower range of sinking rates observed in the field (all parameters are listed in Table 2). Chemical and biological interactions
are employed in the interaction term RC, which is different for each variable (C) based on relevant processes.

The rate of change in the iSPM concentration ($C_t$ term) is calculated as Eq. (3):

$$\frac{dC_{iSPM}}{dt} = C_{iSPM} + C_{iSPMinput},$$ (3)





The interaction term $R_C$ is calculated as Eq. (4):

$$R_{iSPM} = [(\lambda_{s2d}C_{sediSPM} - \lambda_{d2s}C_{iSPM})/dz]_{z=bottom}, \tag{4}$$

iSPM enters a new sediment pool with a sedimentation rate ($\lambda_{d2s}$) of 3.5 m·day$^{-1}$ if bottom stress<$\tau_{crit}$ and a resuspension rate

($\lambda_{s2d}$) of 26 day$^{-1}$ if bottom stress> $\tau_{crit}$. Critical bottom shear stress ($\tau_{crit}$) was set to 0.07 N·m$^{-2}$, which is in a range reported by

Wölfl et al. (2014).

As sed$_{iSPM}$ exchanges occur locally at the bottom and the group is not exposed to mechanical displacement, Eq. (2) is

simplified as Eq. (5):

$$\frac{dC_{sediSPM}}{dt} = [R_{sediSPM}]_{z=bottom}, \tag{5}$$

The interaction term $R_C$ is calculated as Eq. (6):

$$R_{sediSPM} = -\lambda_{s2d}C_{sediSPM} + \lambda_{d2s}C_{iSPM}, \tag{6}$$

As the iSPM has an impact on light penetration, the photosynthetically active radiation in the water column has been updated

and is calculated as Eq. (7):

$$I(x,y,z,t) = \frac{I_S(x,y)}{2}exp\left(-k_w z - k_{Chl}\int_z^0 \sum_{j=1}^2 Chl_{Pj}\partial z - k_{iSPM}\int_z^0 C_{iSPM}\partial z - k_{DOM}\int_z^0 C_{DOM}\partial z\right), \tag{7}$$

where $I_S(x,y)$ is short wave radiation (W·m$^{-2}$) at the surface, x and y identify the horizontal grid points, z is the water depth

in m, and $k_x$ are extinction coefficients (Table 2).

In Hornsund, most of the variability of the optical properties in the summers of 2009 and 2010 was attributed to particles of

mineral origin (Sagan and Darecki, 2018) and thus the input of organic particles with meltwater was considered negligible

here. The attenuation coefficient specific for iSPM measured in another polar fjord in Greenland (0.13 m$^2$·g$^{-1}$) (Lund-Hansen

et al., 2010) was high compared to other published values: 0.07 m$^2$·g$^{-1}$ (Christian and Sheng, 2003), 0.06 m$^2$·g$^{-1}$

(Pfannkuche and Schmidt, 2003), 0.065 m$^2$·g$^{-1}$ (Oliver et al., 2020). Thus, here 0.065 m$^2$·g$^{-1}$ light extinction coefficient

($k_{iSPM}$) was prescribed to the model, which gave reasonable results in terms of light limitation and is in the range of field

measurements.

The light limitation also depends on the plankton photosynthesis efficiency parameter (*a*). Here, it was increased to 0.04

(W·m$^{-2}$)$^{-1}$, which is within the range reported for Arctic coastal and shelf waters (Van De Poll et al., 2018; Stuart et al., 2000;

Strom et al., 2016; Platt et al., 1982) and is in line with previous studies showing that fjord plankton communities are

adapted to low light (Simo-Matchim et al., 2016; Holding et al., 2019). The light limitation is calculated as Eq. (8):

$$\alpha(I) = tanh(a)I(x,y,z,t), \tag{8}$$



In this 1D setup, we do not simulate fish due to their migration, which reduces the uncertainty of the current simulations. Thus, the macrobenthos loss term only consists of excretion ($\varepsilon_{MB}C_{MB}$), and natural mortality ($m_{MB}C_{MB}$) as in Eq. (9):

$$R_{MB_{loss}} = \varepsilon_{MB}C_{MB} + m_{MB}C_{MB}, \tag{9}$$

Similarly, the reaction terms for zooplankton, detritus and DOM were changed accordingly to remove fish grazing (Daewel et al., 2018). Predation mortality from the fish functional group was accounted for by increasing macrobenthos natural mortality ($m_{MB}$) to 0.03 day$^{-1}$.

We do not provide nutrient input with meltwater due to the lack of data for parametrisation and to disentangle it from the effect of iSPM discharge; thus, the burial rate in the carbon and nitrogen sediment pool (sedCN, Eq. 10) and Si (Eq. 11) is set to 0 to prevent decreasing nutrient concentrations over the simulation time. For the full description of the equations, the reader is referred to (Daewel and Schrum, 2013). We speculate that the bias introduced by not providing nutrient input is relatively low considering the characteristics of the discharge (see 4.5).

$$R_{sedCN} = \lambda_{d2s}C_D - \lambda_{s2d}C_{sedCN} - \theta(O_2)2\varepsilon_{sedCN}(T)C_{sedCN} - \theta(-O_2)\varepsilon_{sedCNdenit}(T)C_{sedCN} - \delta_{bur}C_{sedCN}, \tag{10}$$

$$R_{sedSi} = \lambda_{d2s}C_{opal} - \lambda_{s2d}C_{sedSi} - \delta_{bur}C_{sedSi}, \tag{11}$$

The carbon burial potential (CB, Eq. 12) was calculated as 70% burial efficiency of the carbon and nitrogen sediment accumulation rate (Koziorowska et al., 2018):

$$CB = \eta_{bur}R_{sedCN}, \tag{12}$$

## 2.4    Model assessment

The satellite data products of suspended particulate matter were not available for the glacial bays and the analysis of the long-term trends and model validation (2005-2009) were not possible here. Thus, we performed the model assessment based on the available field data from 2015-2021. The summertime mean iSPM concentration for the past conditions was reconstructed based on measurements of iSPM concentration and PDD AT, and it showed a high correlation with simulated iSPM concentration parametrized based on the complementary dataset of sediment flux measurements (Sup. Fig. 3, $R^2$ = 0.928, p = 0.009). The iSPM concentration at modelled station 2 (HH1) in 2006 and 2009 was also compared with the iSPM field data at monitoring stations M4 (H1_09) and M5 (H1_11) from 2019 (Sup. Fig. 2), which represented environmental conditions (PDD SST, PDD AT, melt season duration, and precipitation in Fig. 3) the closest to the simulation period. Results showed that the model realistically simulated the seasonal pattern and vertical distribution of the iSPM. Despite the fact that the iSPM input was parametrized for Hansbukta, which was the only bay with sufficient data and most studied in Hornsund, and the iSPM load and discharge can differ between glaciers, the spatial patterns from measurements of iSPM at the surface conducted in all Hornsund in summer 2017 were in line with the simulation results (Sup. Fig. 4). The literature



data (Sup. Tab. 4) of concentrations of all the nutrients and functional groups showed that the model performed well when compared to the current knowledge of the West Spitsbergen fjords.

## 2.5    Data analysis and visualization

The maps and satellite images were generated and processed in ArcGIS Pro 2.8.0. The plots were prepared in Python 3.7 (Van Rossum and Drake, 2009) using Matplotlib 3.1.1 (Caswell et al., 2019), Pandas 1.0.5 (Mckinney, 2010; Reback et al., 2020), and seaborn 0.11.1, and arranged in Inkscape 0.92.4.

The Hamed and Rao modified Mann-Kendall (mMK) test was used to determine whether a trend exists in time series data (SIF, PDD SST, PDD AT, iSPM, precipitation) with a significance level of 0.05 (*) and 0.001 (**) (Python library pymannkendall 1.4.2).

For each modelled station and each scenario, the 5-years (2005-2009) averages of SIC and SIT in May, mean summertime integrated iSPM, and rates of phytoplankton primary production (phyPP), zooplankton secondary production (zooSP), and carbon burial (CB) were calculated. Then, the average values of phyPP, zooSP, and CB rates for all 20 stations were multiplied by the average newly ice-free area between 2006 and 2010 (64.21 km$^2$). The resulting phyPP, zooSP and CB under the SPM scenario were considered as gains in carbon sequestration potential due to the marine-terminating glaciers retreat, whereas the differences between noSPM and SPM scenario were considered as losses due to the iSPM discharge with meltwater.

The influence of iSPM discharge on the ecosystem dynamics was exemplified by presenting biomass of ice algae (IA) and macrobenthos (MB), as well as biomass of phytoplankton (PHY), zooplankton (ZOO), silicate and light limitation index (SIL, LLI) integrated for the whole water column at three modelled stations (2, 9, 14) that were comparable due to similar depths (42.45 – 49.55 m), but presented low, intermediate and high level of summertime iSPM input. Also, two years with contrasting sea ice conditions (2008 and 2009) were displayed.

## 3    Results

### 3.1    Newly ice-free marine habitats

The area of newly ice-free coastal waters due to the retreat of marine-terminating glaciers in Hornsund increased by ~99.4 km$^2$ between the summers of 1976 and 2022 (Fig. 1, 3, around 38% increase in the total area), whereas the volume gained until 2010 was ~3.3 km$^3$. The trends were linear (y = 2.1406x - 4231.2; R² = 0.995 for the area, and y = 0.097x - 191.38; R² = 0.984 for volume, t-test p<0.001) with rates of ~21.4 km$^2$·decade$^{-1}$ and 1.0 km$^3$·decade$^{-1}$ (Fig. 3). While advances in glacier fronts due to surge events were observed for some marine-terminating glaciers in Hornsund, these did not influence the overall increasing trends. Along with the glacial retreat, the number of SIF days (fraction of the year with SIC<15%) increased significantly (~0.1 decade$^{-1}$, i.e. around 44 days, p<0.001 mMK test). Despite high interannual variability, the



central part of Hornsund has become mostly devoid of sea ice since 2006, but there still is seasonal sea ice cover in the newly formed glacial bays (Fig. 6a).

## 3.2    Melt and SPM discharge potential

The annual sum of daily SST> 0°C (PDD SST), showed no significant trend in outer Hornsund due to strong variability between years (p>0.1 mMK test) (Fig. 3), but it was significantly increasing for summer months (June – August;
46.8°C·decade⁻¹, p<0.05 mMK test). The annual and summertime sum of positive daily air temperatures (PDD AT), as well as annual precipitation, showed significant increases (60.5°C·decade⁻¹, 31.4°C·decade⁻¹, and 56.0 mm·decade⁻¹, respectively, p<0.001 mMK test). Melt season duration increased significantly (p<0.001 mMK test) with a rate of ~9 day·decade⁻¹ (2.5% of the year). At the beginning of the measurements, the melt season started in June and ended in late September – mid-October, whereas currently it can start as early as February and ends mostly in October (Sup. Fig. 5).

The 6-year monitoring dataset of summertime SPM concentration in Hansbukta (Fig. 1c) was not sufficient to show long-term trends. However, average integrated iSPM levels were correlated with both the annual sum of PDD AT (y = 0.061x - 19.549, R² = 0.68, p<0.05 t-test) and the summertime sum of PDD AT (June to August) (y = 0.221x - 75.047, R² = 0.78, p<0.05 t-test). Even though the correlation was stronger for the summertime PDD AT, the estimates displayed numerous negative values. However, the annual sum of PDD AT allowed coarse reconstruction of past conditions and revealed
significant increases in iSPM concentration (3.7 g·m⁻³·decade⁻¹ in 1979-2022, p<0.001 mMK test). Importantly, within the modelled time range (2005-2009, Fig. 3, grey shade), both iSPM estimates gave similar results in 2006 and 2009 (8.6 and 12.0; 8.1 and 9.8 g·m⁻³, respectively).

## 3.3    SPM dynamics

The concentration of iSPM varied between seasons with the highest levels in July – October (up to 150 g·m⁻³) and the lowest
between November and May (up to 50 g·m⁻³), whereas the highest levels of organic SPM were observed between April and June (up to 20 g·m⁻³) (Fig. 4a). Sediment flux observed for iSPM ranged between 1 – 6648 g·m²·day⁻¹ while for organic SPM it was 0.9 – 333 g·m²·day⁻¹ (Fig. 4b). The sinking rate of iSPM ranged between 0.6 – 265 m·day⁻¹ (mean 25.3, median 12.2 m·day⁻¹) (Fig. 4c), while the sinking rate of organic SPM was one order of magnitude lower with a range of 0.3 – 28.9 m·day⁻¹ (mean 2.8, median 1.7 m·day⁻¹). The sediment flux of iSPM, which represents temporary dynamics of iSPM input,
was dependent on the accumulated daily air temperature for positive degree days for a 6-day window (6accPDD AT) and mean salinity in the layer above (R² = 0.662, p<0.001 t-test, Fig. 4d). Within the range of frequently observed values of 6accPDD AT (0 – 40°C) and salinity (30 – 35) the estimated inorganic sediment flux could reach up to 1860 g·m²·day⁻¹ (Sup. Fig. 1). Importantly, the regression model (Eq. 1) performed well for inorganic sediment flux<2000 g·m²·day⁻¹ (RMSD = 290.1 g·m²·day⁻¹), which consisted 95% of the dataset, and mostly underestimated the highest inorganic sediment flux
values (RMSD = 823.3 g·m²·day⁻¹ for all the dataset).





### 3.4 Spatial patterns of sea-ice, iSPM, plankton production and carbon burial

The mean SIT and SIC in May were the highest in the southern and inner parts of Hornsund (5-year average up to 8 cm and 19.3%, respectively) and the lowest in the northern and outer parts (5-year average of 3 cm and 8.7%, respectively) (Fig. 5a). The mean summertime integrated iSPM concentration was the highest in the inner glacial bay (modelled station 14; 5-year average: 164.4 g·m$^{-3}$), where rates of plankton primary and secondary production, and carbon burial were the lowest (5-year average: 11.0, 1.5, and 5.5 gC·m$^{-2}$·y$^{-1}$, respectively; Fig. 5b,c,d). At other stations, the mean summertime integrated iSPM concentration was in a range between 2.1 – 7.1 g·m$^{-3}$ which allowed phyPP to reach rates between 66.3 – 100.7 gC·m$^{-2}$·y$^{-1}$ (versus 131.3 – 171.2 gC·m$^{-2}$·y$^{-1}$ under noSPM scenario), whereas rates of zooSP were between 17.7 – 47.2 gC·m$^{-2}$·y$^{-1}$ (versus 48.7 – 75.7 gC·m$^{-2}$·y$^{-1}$ under noSPM scenario), and CB rate was in a range of 6.0 – 17.7 gC·m$^{-2}$·y$^{-1}$ (versus 6.5 – 23.0 gC·m$^{-2}$·y$^{-1}$ under noSPM). In the simulation period (2005-2009), the newly ice-free areas in Hornsund substantially contributed to phyPP, zooSP, and CB (on average 5.1, 2.0, and 0.9 GgC·y$^{-1}$, respectively – Fig. 5, green, gains in carbon sequestration potential in SPM scenario). However, the potential was hindered by iSPM input by 5.0, 2.1 and 0.1 GgC·y$^{-1}$, respectively (Fig. 5, red, loss due to the difference between noSPM and SPM scenario). Thus, without the release of mineral particles, plankton primary and secondary production could have been around two times higher (10.1, 4.1 GgC·y$^{-1}$ under noSPM scenario), whereas carbon burial was less affected by iSPM input (1.0 GgC·y$^{-1}$ under the noSPM scenario, around 16.5% higher than carbon burial under SPM scenario).

### 3.5 Ecosystem dynamics

The ecosystem dynamics related to the sea ice and iSPM in the newly ice-free areas was presented for three modelled stations with low, intermediate and high influence of iSPM (stations 9, 2, and 14, Fig. 6abc, respectively) in two contrasting years (cold 2008 and warm 2009). The sea ice thickness and concentration (SIT and SIC) were lower in 2008 than in 2009, and in the outer than in the inner glacial bay. Thus, only in 2009 did the ice algae bloom reach up to 0.16 gCm$^{-2}$ biomass in inner Hornsund (9, 14 in Fig. 6a,c) and sea-ice presence (up to 0.5 m) delayed the phytoplankton bloom by around 10 days. Under the low and intermediate influence of iSPM (Fig. 6a,b), the light limitation index was slightly lowered before the main melt season (March to early June) and the significant effect of light limitation due to iSPM input started around late June (up to 24 and 6 gm$^{-3}$ at stations 9, 2 in Fig. 6a,b). Due to the worsened underwater light conditions, the peaks of spring and summer phytoplankton blooms were delayed around 10 – 14 days and the summer peak reached lower biomass (~0.4 – 0.5 gC·m$^{-3}$ under the SPM scenario, and 0.7 gC·m$^{-3}$ under noSPM scenario), which further affected zooplankton (peak delayed by ~9 days and 0.1 – 0.2 gC·m$^{-3}$ less biomass) and macrobenthos (~10 gC·m$^{-2}$ less biomass). At the highest levels of iSPM (up to 500 g·m$^{-3}$ at station 14, Fig. 6c), strong light limitation started early in March. Thus, phytoplankton, zooplankton, and macrobenthos reached very low biomass (<0.2 gC·m$^{-3}$, <0.05 gC·m$^{-3}$, and <5 gC·m$^{-2}$, respectively). The delays in phytoplankton bloom related to the iSPM led to delays in silicate limitation and increases in the ice algae biomass in spring, particularly at the station with the highest levels of iSPM (up to 0.01 gC·m$^{-2}$ difference between SPM and noSPM scenario).



## 4   Discussion

### 4.1   Newly ice-free marine habitats

340   We report significant increases in new marine habitat area (~100 km$^2$) and volume (>3.3 km$^3$) between 1976-2022 in Hornsund (Fig. 1b, 3, 7a) due to the retreat of marine-terminating glaciers. These results are in line with cryosphere studies in West Spitsbergen fjords (Błaszczyk et al., 2021, 2023; Grabiec et al., 2018; Strzelecki et al., 2020) and in polar regions in general (Kochtitzky et al., 2022; Pfeffer et al., 2014). In the coastal Arctic and Antarctic, glaciers and ice sheets have lost mass due to the increased submarine (basal) melting and iceberg calving (dos Santos et al., 2021; Błaszczyk et al., 2013, 2023), and in Svalbard, a doubling of ice mass loss was predicted by 2100 (Geyman et al., 2022). The retreat of many marine-terminating glaciers has already produced newly ice-free areas, and some of them have receded onto land (Błaszczyk et al., 2013; Jerosch et al., 2019; Kochtitzky et al., 2022). Recently, the rapid loss of numerous glaciers was related to both external forcing such as increases in atmospheric and oceanic temperatures and lack of sea-ice buttressing or internal dynamics such as surges (Błaszczyk et al., 2013, 2023; Strzelecki et al., 2020). Here, we show increasing trends in the length of the melt season (~9 day·decade$^{-1}$) and the sum of PDD (46.79°C·decade$^{-1}$ for summer PDD SST, 60.54°C·decade$^{-1}$ and 31.43°C·decade$^{-1}$ for annual and summer PDD AT, respectively, Fig. 3), and thus a rising potential for melting. This finding is potentially important for predictions in other regions such as Greenland, Patagonia, Alaska, and the Antarctic Peninsula, which experience temperatures close to or above the melting point and hence are exposed to similar warming effects.

Furthermore, we report a significant loss in sea ice duration in central Hornsund (~44 day·decade$^{-1}$) (Fig. 3). However, as glacial retreat opens new coastal areas, it also increases the potential for winter sea ice formation in the more protected inner bays (Fig. 5a). In contrast to the glaciers which mass loss cannot be stopped nor reversed once induced, sea ice was shown to be more responsive to variations of both ocean and air temperatures (Muckenhuber et al., 2016). Thus, there still can be land-fast ice (sea ice attached to the coastline) covering the inner parts of West Spitsbergen fjords for a limited time during winter and spring. Moreover, the ice bridge in inner Hornsund (Fig. 1b, 7a) is predicted to vanish in the coming decades (2030–2055) (Grabiec et al., 2018; Osika et al., 2022), which will transform Hornsund from a fjord into a strait enabling sea ice advection from the Barents Sea. However, the loss of the ice bridge could also result in the increased presence of warm Atlantic Water in the area, and therefore, further sea ice loss. These seemingly contrasting predictions highlight the importance of continuous evaluation of the changing Hornsund environment and its potential as a model area for studies on regime shifts.

### 4.2   SPM dynamics

Based on the coarse reconstruction and modelling results presented in this study, we suggest that the Hornsund bays have already been under the strong influence of dark glacial plumes since the beginning of the simulation period (2005) (Fig. 3, 5ab, 6). iSPM concentration increased after 2013 and further rises are expected (3.7 g·m$^{-3}$·decade$^{-1}$ integrated for the water column in summer). We show that air temperature variability, specifically the accumulated daily air temperature above the





melting point for 6-day window (6accPDD AT), which takes into account the delays in meltwater discharge, modulates the iSPM flux (Fig. 4d, Sup. Fig. 1), similarly as was suggested for a glacial bay in Kongsfjorden – another West Spitsbergen fjord (D'Angelo et al., 2018). Recent studies also indicated that sediment production and fluxes to the coastal zones in the polar regions have increased due to higher air temperatures (Overeem et al., 2017; Szczuciński et al., 2009). Thus, it is anticipated that even central fjords will receive high input of mineral particles in the future as turbid glacial plumes will spread farther from the source (Fig. 7a) (Castelao et al., 2019; Kanna et al., 2018; Hudson et al., 2014), therefore extending the influence of meltwater discharge onto the shelf and considerably affecting marine systems downstream (Meire et al., 2017, 2015; Milner et al., 2017).

The iSPM discharge was the most extensive during summer (Fig. 4), although it could also be observed during autumn and winter, when it is intensified by tidal resuspension, resulting in a relatively high concentration of organic and inorganic suspended particles (Moskalik et al., 2018). In the future, more days with open-water conditions (no sea ice), which can increase wave action and particle removal from the beaches and tidal flats, as well as a longer melt season (Fig. 3, Sup. Fig. 5) could potentially lead to iSPM affecting a substantial part of the productive season, including not only summer and autumn but also spring. Here, we show high variability of iSPM dynamics with sinking rates between $0.6 – 265.9$ m·day$^{-1}$ and sediment flux between $1.0 – 6647.7$ g·m$^{-2}$·day$^{-1}$ (Fig. 4bc) which should be investigated further in the context of the driving mechanisms such as flocculation (Moskalik et al., 2018).

### 4.3 Ecosystem dynamics

Observational data and previous modelling studies showed that the continuing retreat of marine-terminating glaciers will negatively affect planktic and benthic communities, especially in enclosed shallow bays such as Brepollen (Fig. 7a) (Neder et al., 2022; Szeligowska et al., 2022, 2021; Torsvik et al., 2019). Indeed, we observed decreases in phytoplankton, zooplankton, and macrobenthos biomass, and delays in their peak occurrence close to the glacial fronts (by around 10-14 days as compared to the noSPM scenario, Fig. 5). These decreases were related to the input of particulate matter from land which, even in relatively low concentration in spring, can affect phytoplankton due to light attenuation (Fig. 7b). Under the SPM scenario, plankton primary production rates reached $66.3 – 100.7$ gC·m$^{-2}$·y$^{-1}$ with the mean summertime integrated iSPM concentration of $2.1 – 7.1$ g·m$^{-3}$, whereas it was around two times higher in the noSPM scenario ($131.3 – 171.2$ gC·m$^{-2}$·y$^{-1}$). Both ranges are comparable with the field measurements in inner and outer Hornsund, and other West Spitsbergen fjords (Hodal et al., 2012; Iversen and Seuthe, 2011; Piwosz et al., 2009; Vonnahme et al., 2021) (Sup. Tab. 4). Sea ice algae biomass was extremely low in most years ($<12$ mgC·m$^{-2}$, except for 2009 – up to 160 mgC·m$^{-2}$) due to thin ice ($<50$ cm) that disappeared before the main productive season. Ice algae did not seem to be negatively affected by iSPM and could likely be favoured in the SPM scenario due to the delays in phytoplankton bloom and thus also in silicate limitation. Importantly, we suggest that sea ice loss leading to the earlier offset of spring pelagic production might become a compensation mechanism for higher iSPM input in summer (Fig. 7b).





The modelled carbon burial rate was within the reported values (Koziorowska et al., 2018; Kuliński et al., 2014; Zaborska et al., 2018) and it constituted around 10–20% of the primary production, which is also in line with the current observations in polar and sub-polar fjords (Włodarska-Kowalczuk et al., 2019). Unfortunately, no field data for the assessment of plankton

secondary production rates exists from this region. However, the values simulated here fell between the plankton primary production and carbon burial rates as expected. Plankton secondary production was reduced due to decreased food base (17.7– 47.2 $gC \cdot m^{-2}y^{-1}$ versus 48.7 – 75.7 $gCm^{-2}y^{-1}$ under the SPM and noSPM scenario, respectively). According to our simulations, carbon burial was the least affected by iSPM (6.0 – 17.7 $gC \cdot m^{-2} \cdot y^{-1}$ versus 6.5 –23.0 $gC \cdot m^{-2} \cdot y^{-1}$ under SPM and noSPM scenarios, respectively). Since the burial of accumulated material depends on the vertical flux of the organic matter

originating from phytoplankton and zooplankton, food intake by benthic fauna, and rates of benthic mineralisation, we hypothesise that the changes in the phytoplankton bloom timing might have shifted the carbon pathway from zooplankton and macrobenthos pool to carbon burial in sediments, and thus carbon burial was still relatively high in the SPM scenario (~16% lower than under noSPM scenario). Only the extremely high levels of iSPM (mean summertime integrated iSPM concentration of 164.4 $g \cdot m^{-3}$), which can be observed directly inside the turbid plumes, resulted in an almost complete

absence of phyto- and zooplankton, and macrobenthos, and in relatively low plankton production rates (11.0 and 1.5 $gC \cdot m^{-2} \cdot y^{-1}$ for phyPP and zooSP, respectively), but still considerable burial rates (5.5 $gC \cdot m^{-2} \cdot y^{-1}$). Thus, we speculate that sediment discharge to polar coastal zones might result in less complex food webs, constituted by species better adapted to high iSPM concentrations and sedimentation rates. It could reduce the biomass that is utilised in the pelagic and benthic system leading to higher carbon burial in sediments (Fig. 7b).

**4.4    Carbon gains and losses**

Marine sediments in polar fjords have recently been recognised as efficient organic carbon sinks and incorporated into global carbon burial estimates (Bianchi et al., 2020; Cui et al., 2022; Smith et al., 2015) highlighting its societal importance as a climate regulation ecosystem service (Barnes et al., 2021; Bax et al., 2021). They might become more efficient in the capture-to-long-term carbon sequestration due to high sedimentation rates and their restrictive nature compared to more open

coastal environments, particularly with the expansion of the shallow and isolated bays and increased land-ocean connectivity (Fig. 7b) (Smith et al., 2015). Here, we show that newly ice-free areas in Hornsund (~64 $km^2$ between 2006–2010) markedly contributed to plankton primary (5.1 $GgC \cdot y^{-1}$) and secondary production (2.0 $GgC \cdot y^{-1}$), and carbon burial (0.9 $GgC \cdot y^{-1}$) (greens in Fig. 6, carbon gains under SPM scenario). This carbon burial constitutes only a small fraction of the globally estimated rates for seafloor ($2.9 \cdot 10^4$ – $1.6 \cdot 10^5$ $GgC \cdot y^{-1}$; Bauer et al., 2013; Cai, 2011; Hedges and Keil, 1995). However,

emerging marine habitats could gain more relevance considering that organic carbon burial efficiency in fjords is two times higher than the global ocean average (Smith et al., 2015) and recognising the scale of marine ice loss across the Arctic and Antarctic. Due to the anticipated negative effects of ice loss (Hunter, 2022), here we show that part of the potential gains in carbon sequestration related to the newly ice-free areas turns into losses for plankton primary (–5.0 $GgC \cdot y^{-1}$) and secondary





production ($-2.1$ GgC·y$^{-1}$), and burial ($-0.1$ GgC·y$^{-1}$) under the SPM scenario (Fig. 6, red). Without the delivery of mineral

particles from land, plankton primary and secondary production could have been around two times higher (10.1 and 4.1 GgC·y$^{-1}$ under noSPM scenario, respectively), whereas carbon burial was less affected by iSPM input (1.0 GgC·y$^{-1}$ under noSPM scenario).

### 4.5 Current limitations and future perspectives

While the coupled physical-biogeochemical model with newly implemented iSPM input performed well according to our

assessment, the field data for model parametrisation and validation were not available for the simulated period (2005–2009), whereas remotely-sensed products for iSPM concentration did not cover the inner fjords and were frequently limited by clouds. Despite that, the reconstructions of previous conditions and assessment based on the two complementary datasets collected in recent years (2015–2021, SPM and sediment flux) suggest that the simulated spatial and temporal dynamics of both inorganic and organic SPM was rather realistic and in line with the current knowledge of the West Spitsbergen marine

ecosystem. While it should be considered that this reconstruction was based on a few years of measurements, which might limit its robustness, particularly towards the beginning of AT measurements, the correlation with annual PDD AT seems to yield reasonable estimates. A recent multi-year study (2010-2016) in another West Spitsbergen fjord (Kongsfjorden) also indicated the relationship between particle fluxes and air temperature above the melting point (D'Angelo et al., 2018). Importantly, the satellite data products calibrated for the glacial bays should become available (Klein et al., 2021; Walch et

al., 2022) to verify the long-term trends in the iSPM discharge revealed in this study.

So far, only a few numerical models have been implemented in the polar and subpolar regions to study the dynamics of SPM input from land. 3D models have indicated the areas with long residence time and high accumulation rates of iSPM (Neder et al., 2022) and considered a light limitation that led to the shallowing of the photic zone within the dark plumes (Le Fouest et al., 2010; Marín et al., 2013; Møller et al., 2023). Moreover, 2D models have been developed to simulate the sedimentation

induced by ice-rafted debris (Mugford and Dowdeswell, 2010), and by glacial meltwater plumes emerging from marine-terminating glaciers (Dowdeswell et al., 2015; Mugford and Dowdeswell, 2011). However, they differ in the parametrisation and approach, both between each other and our study, due to the various data, processes represented, and numerical models available for the respective regions. To the best of our knowledge, this is one of the first attempts to implement the influence of iSPM in the coupled physical-biogeochemical model in polar coastal zones. While this 1D approach does not represent

upwelling or spatial fluctuations in the glacial plumes, e.g. implemented in Hansbukta in 2D (De Andrés et al., 2018, 2021), or flocculation of the particles (Dowdeswell et al., 2015; Mugford and Dowdeswell, 2011), it is a first step to address the technical challenges related to the coupling between the sympagic, pelagic and benthic systems and their response to glacial discharge and retreat.

Even though Hornsund is amongst the most-studied Svalbard fjords, our study was limited to 5 years period due to the lack

of long-term input data for temperature and salinity (Jakacki et al., 2017; Torsvik et al., 2019) as most of the hydrodynamic models do not simulate coastal zones with sufficient horizontal resolution and they do not consider changes in the glacial





bays' extent. It should also be considered that sea ice concentration and thickness were extracted from the closest data points available, and thus sea ice conditions might have been different in the glacial bays. However, smoothing the data for more stable model runs could have resulted in more accurate forcing. Also, the advection of Atlantic Water is not represented in

this 1D setup, but due to the strong boundary in the form of polar front and sills, most of the primary and secondary production in glacial bays of Hornsund is assumed to be local, contrary to other West Spitsbergen fjords experiencing high advection of plankton (Basedow et al., 2004; Gluchowska et al., 2016).

Here, we disentangled the effects of iSPM input from other factors such as organic matter and nutrient delivery with meltwater. The influence of terrestrial organic matter on light attenuation was assumed negligible in Hornsund for the time

of simulation (Sagan and Darecki, 2018). Despite that, the release of large amounts of petrogenic organic carbon that has been isolated for millennia under the ice is recently emerging as an important component of the carbon burial in fjords and its fluxes as well as transformations by microorganisms, which lead to a greenhouse gas emissions, should be better constrained for the future model development (Fig. 7b) (Ruben et al., 2023). Moreover, several modelling and field studies in Arctic coastal waters have shown that the upwelling effect of submarine plumes and nutrient fluxes with meltwater

supports primary production in the glacial bays and on the shelf (Castelao et al., 2019; Luo et al., 2016; McGovern et al., 2020; Oliver et al., 2020). Yet, the net effect depends on the lithology, subglacial discharge rate and depth of the glacier grounding line, as well as the seasonal dynamics of coastal currents, winds, and eddy activity, and it was not possible to represent it properly in this study. Studies in Greenland fjords indicate that macronutrients were primarily supplied to the surface waters by mixing and not the transport from land with glacial meltwater as it was shown to have a relatively low

nutrient load (Hopwood et al., 2020). Furthermore, while macronutrient concentrations can be higher in the Arctic rivers, most of the discharge in Hornsund comes from marine-terminating glaciers (Błaszczyk et al., 2019). Also, rivers were shown to deliver nutrients mostly in August (McGovern et al., 2020), when phytoplankton is already limited due to the light attenuation by iSPM. Thus, the overall bias introduced by not providing nutrient input in our simulations might be relatively low.

The ecosystem dynamics is a result of the combined interaction of, inter alia, dynamic coastline, hydrographic and sea-ice conditions, nutrients and sediment discharge, and thus this interdisciplinary work adds to the current understanding of the complex influence of glaciers on marine productivity and carbon fluxes (Hopwood et al., 2020). The presented numerical framework allows to disentangle the effects of various processes and efficient hypothesis testing. Despite inherent weaknesses, it provides reliable results comparable with the filed measurements. The limitations of this study could be

readily addressed by further development and implementation of high-resolution general circulation models in polar regions (Szeligowska et al., in review) and coupling with biogeochemical modules such as those presented here. Thus, skilful 3D fine-scale ecosystem models could arise from such work in the future.

## 5    Conclusions

In this study, we used Hornsund as a model high-latitude fjord particularly sensitive to a changing climate. We presented the
accumulated effects of interactions between the atmosphere, ocean, cryosphere, and the dynamic coastline, and how these
affect the carbon sequestration potential. By combining the results of numerical modelling, remote sensing, and in situ
observations, we provided a broad view of the periglacial environment and a framework for future simulations of ecosystem
dynamics affected by terrigenous matter input with meltwater. Relatively well-studied areas adjacent to rapidly retreating
marine-terminating glaciers in Hornsund are representative of similar coastal environments and, therefore, shed light on the
formation and development of new marine habitats not only on a local but also on a regional scale. Here, we show that
despite the negative influence of iSPM input, the loss of marine ice in polar regions can be expected to ultimately lead to
higher net productivity and the emergence of carbon sinks due to the formation of newly ice-free areas. However, the
intertwined complexity of changes in high-Arctic coastal zones complicates the estimation of net effects on carbon burial in
sediments. Considerable uncertainties remain, in particular related to the petrogenic organic carbon release. Thus, glacial
retreat and terrigenous matter input should be implemented in current ocean models applied to such coastal systems to
resolve carbon fluxes more accurately. Here, we also highlight the importance of maintaining long-term observations and
implementing FAIR principles (findability, accessibility, interoperability, reusability) in data infrastructures to improve our
understanding of the evolution of deglaciating coasts and subsequent influences on the marine ecosystem, which is one of
the research priorities in the context of climate change impacts on polar regions.

## 515    6    Data availability

The satellite images are available at https://glovis.usgs.gov/app. Meteorological data from Hornsund were downloaded from
https://doi.pangaea.de/10.1594/PANGAEA.909042 and https://doi.org/10.5194/essd-12-805-2020. Datasets for suspended
particulate matter, sediment flux, and salinity were downloaded from https://dataportal.igf.edu.pl/group/longhorn. Arctic Sea
and Ice Surface Temperature datasets were deposited at 10.6084/m9.figshare.24142965 and results of numerical simulations
were stored at 10.6084/m9.figshare.24143013 and 10.6084/m9.figshare.24142992.

## 7    Authors contribution

Contributed to conception and design: MS, DB, BM
Contributed to acquisition of data: MS, AP, DB, MM
Contributed to analysis and interpretation of data: MS, DB, MM, BM
Drafted the article: MS
Revised the article: MS, DB, AP, BM, MM, ET, KBS
Approved the submitted version for publication: MS, DB, AP, BM, MM, ET, KBS



## 8    Competing interests

The authors declare that they have no conflict of interest.

## 9    Acknowledgements

MS, ET, KBS were funded by Polish National Science Centre project (NCN, CoastDark 2018/29/B/NZ8/02463). MS, MM, ET were supported by project financed within the GRIEG competition funded by the Norwegian Financial Mechanism 2014-2021 (No. of agreement: UMO-2019/34/H/ST10/00504). MS was additionally funded by DAAD short-term research grant 2020 (57507441) and NAWA Bekker programme (BPN/BEK/2021/1/00258). DB was supported by Changing Arctic Ocean project MiMeMo (NE/R012679/1) jointly funded by the UKRI Natural Environment Research Council (NERC) and the German Federal Ministry of Education and Research (BMBF/03F0801A). Data was collected in the LONGHORN - oceanographical monitoring realized in Polish Polar Station Hornsund.

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

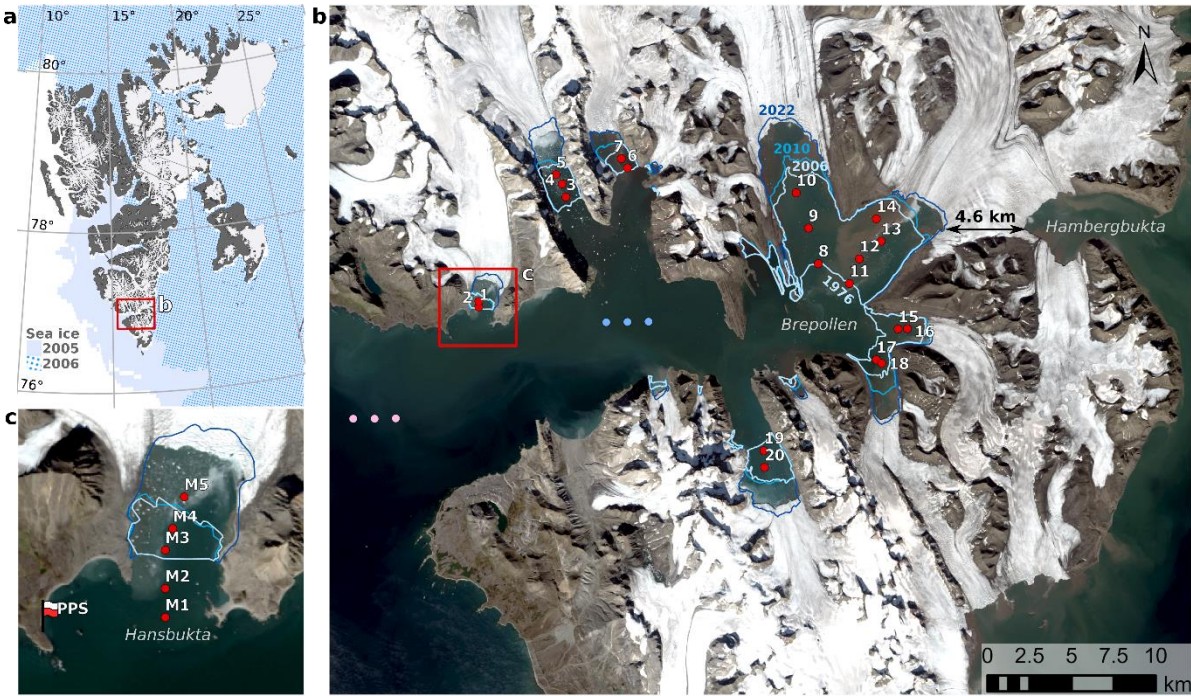

**Figure 1: a) Svalbard archipelago with monthly mean sea-ice extent (SIC>15%) in March 2005 (plain colour) and 2006 (dotted).**

**Land and glaciers extent downloaded from https://geodata.npolar.no/. The red frame indicates the location of Hornsund. b) Newly-ice-free areas in Hornsund which have opened since 1976 (blue lines – glaciers' front position in 1976, 2006, 2010, and 2022) with the width of the ice bridge between Brepollen and Hambergbukta. The dots indicate modelled stations (1-20, red), and three data points for SST (pink) and SI (blue) each. The red frame indicates the location of Hansbukta. c) Long-term SPM and sediment**

**flux monitoring stations in Hansbukta (M1-5, red dots) and Polish Polar Station Hornsund (PPS). Landsat8 satellite image (04/08/2020) downloaded from https://glovis.usgs.gov/app.**



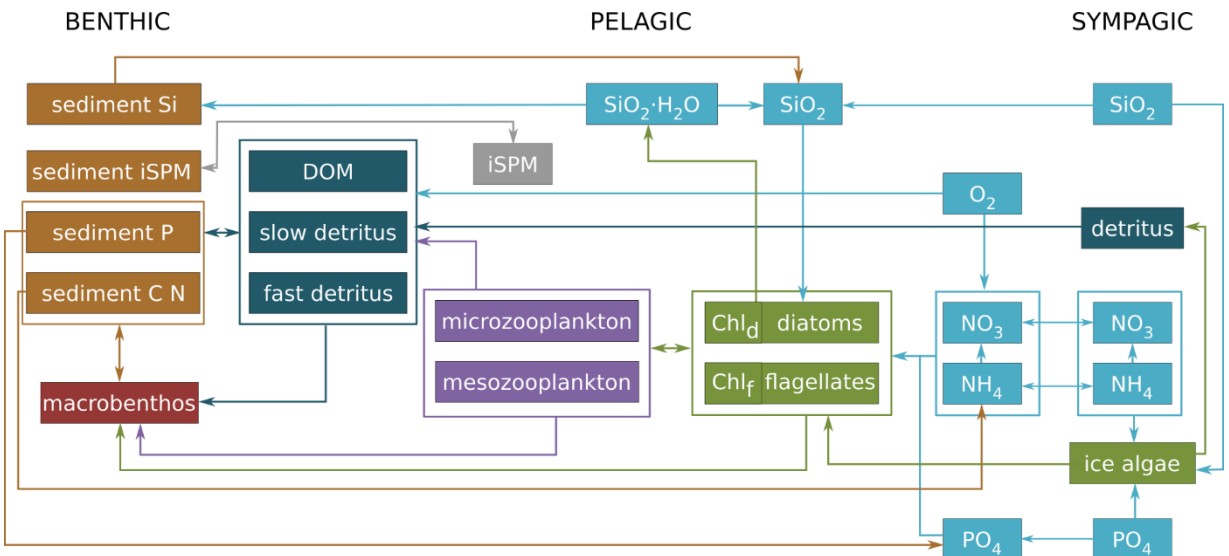

**Figure 2: Schematic diagram of the biogeochemical model ECOSMO-E2E-Polar with new iSPM group. The three systems (benthic, pelagic, and sympagic) are included.**



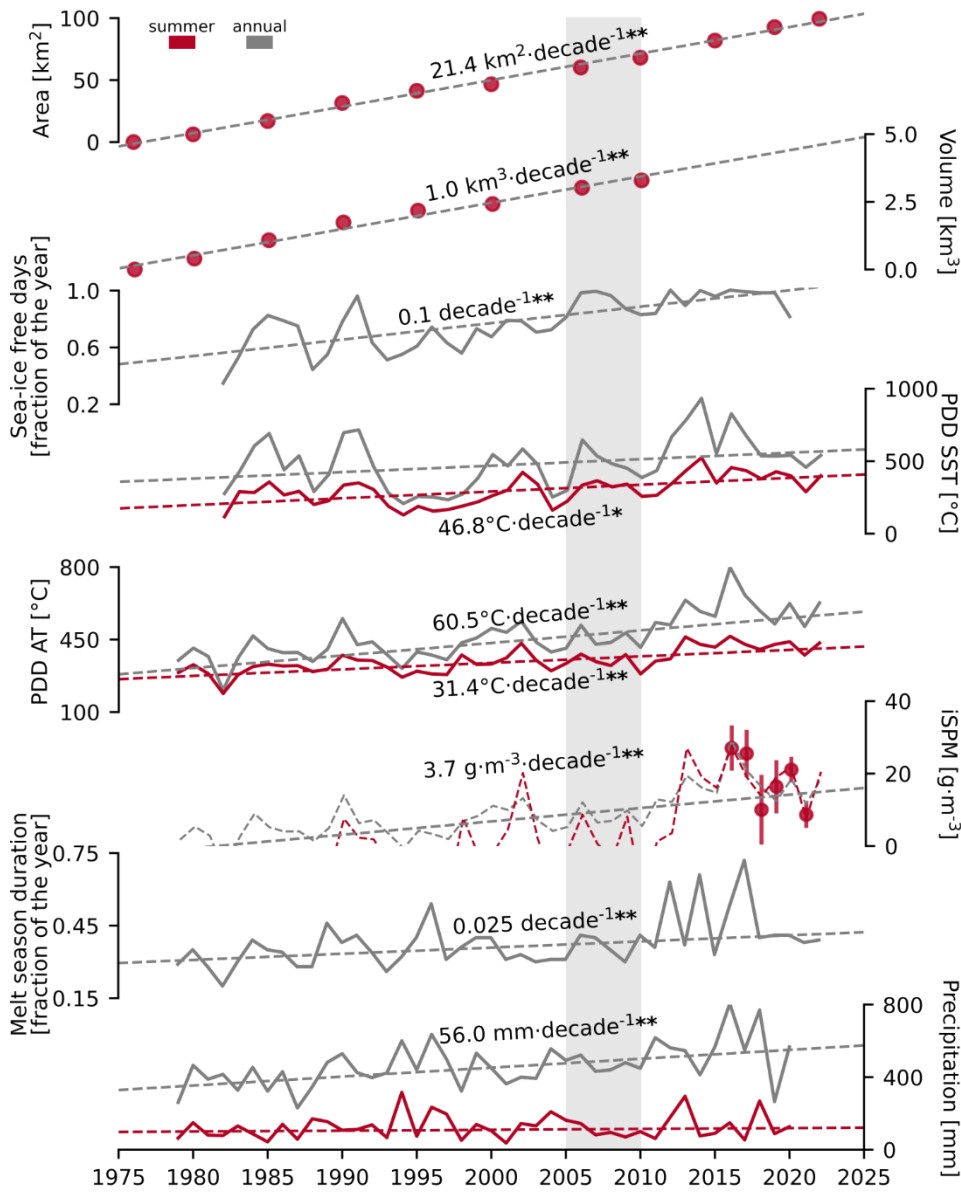


**Figure 3: Long-term trends in the newly ice-free marine habitat and melt potential in Hornsund.** Area (km²) and volume (km³) of newly ice-free marine habitat (assessed for summers between 1976 and 2022), sea ice-free days (fraction of the year, 1982-2020), accumulated positive degree days for sea surface temperature and air (PDD SST and PDD AT, °C, 1979-2022), inorganic SPM concentration reconstructed from 6 years of monitoring (g·m⁻³, 1979-2022), melt season duration (fraction of the year, 1979-2022), and precipitation (mm, 1979-2020). * p<0.05, **p<0.001 for modified Mann-Kendal test. Grey shading indicates the modelling period (2005-2009).



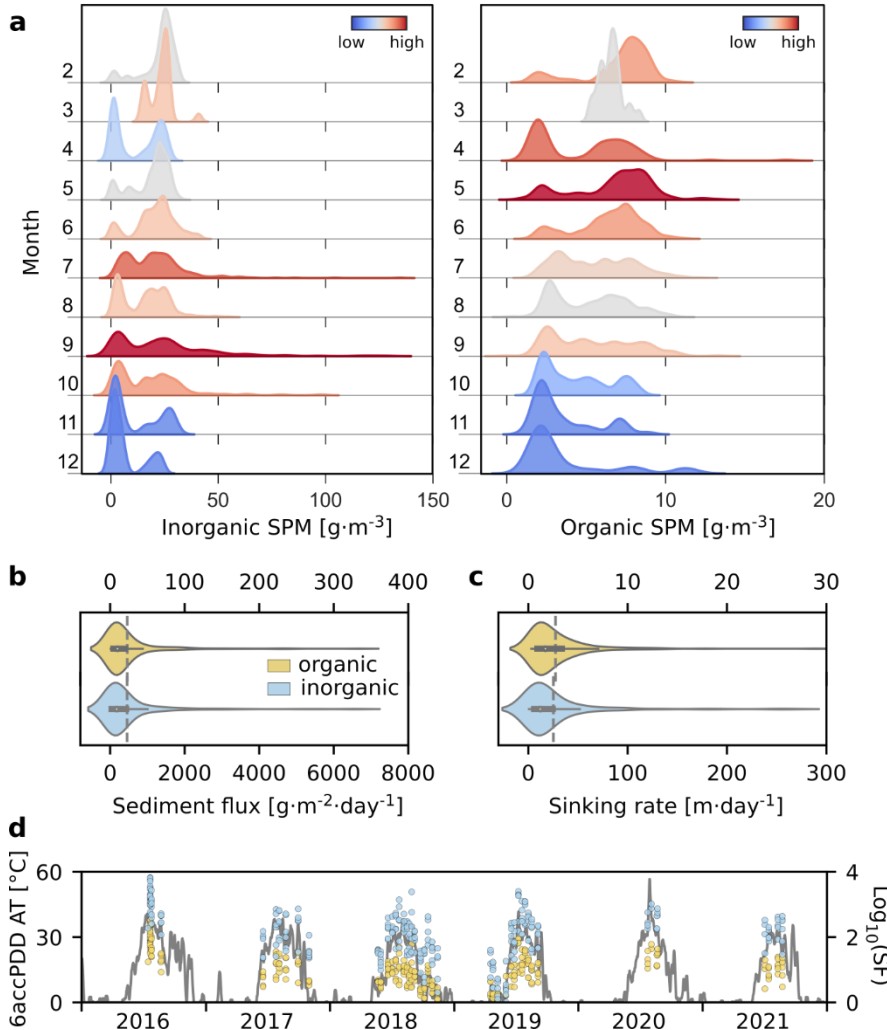

**Figure 4: SPM dynamics at monitoring stations in Hansbutka in 2016-2021. a) Kernel density estimates of SPM concentration (g·m-3, inorganic - left, organic – right). Colours indicate the distribution between months (high to low) b) Inorganic (blue) and organic (yellow) sediment flux (g·m⁻²day⁻¹, grey dashed line - mean value). c) Inorganic and organic matter sinking rate (m·day⁻¹, grey dashed line - mean value. d) inorganic and organic sediment flux (SF, dots, log scale) and accumulated daily air temperature for positive degree days for 6 days window (°C, 6accPDD AT, line).**






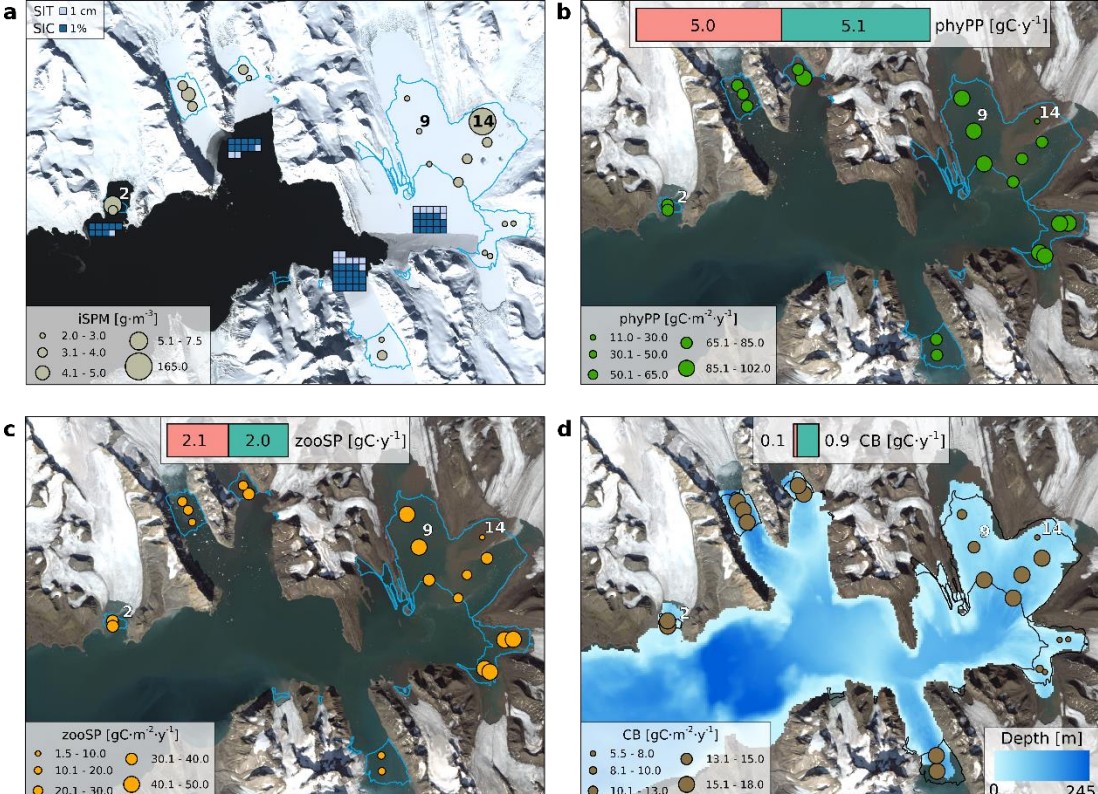

**Figure 5: Spatial patterns in average sea ice thickness (SIT, cm) and concentration (SIC, %) in May, summertime integrated**
**inorganic SPM concentration (iSPM, g·m⁻³) (a), plankton primary (b) and secondary (c) production (phyPP, zooSP), and carbon burial (d, CB) (gC·m⁻²y⁻¹) with blue carbon gains due to the retreat of marine-terminating glaciers (green, SPM scenario) and losses due to the inorganic SPM discharge with meltwater (pink, noSPM-SPM scenario, GgC per year, average for 2005-2009). The lines indicate newly ice-free areas extent in 2006. Ecosystem dynamics at stations 2, 9, and 14 is presented in Fig. 6. Landsat8 satellite images (a - 14/05/2022 and b, c, d - 04/08/2020) downloaded from https://glovis.usgs.gov/app.**






**Figure 6: Changes in ecosystem dynamics due to iSPM input in 2008 and 2009 at three modelled stations: a – low iSPM influence (station 9), b – medium iSPM influence (station 2), c – high iSPM influence (station 14). Line plots show sea ice thickness (SIT, m), the biomass of ice algae (IA, gC·m⁻² or mgC·m⁻², black), integrated silicate (grey) and light limitation index (SLI, LLI, -), integrated biomass of phytoplankton and zooplankton (PHY and ZOO, gC·m⁻³), and macrobenthos biomass (MB, gC·m⁻²). Full line – SPM scenario, dashed line – noSPM scenario. SLI and LLI equal to 1 indicate that phytoplankton is not limited either by silicate or light. Colour plots indicate the SIC<15% (blue, open water), differences in ice algae, phytoplankton, zooplankton and**





macrobenthos biomass between SPM and noSPM scenario (SPM-noSPM), and integrated inorganic SPM concentration in the SPM scenario (iSPM, g·m⁻³). Brown arrows indicate the start of the melt season (30th of May 2008 and 3rd of June 2009) and black arrows indicate delays in peak abundance of phytoplankton and zooplankton. Note the different scales (*).

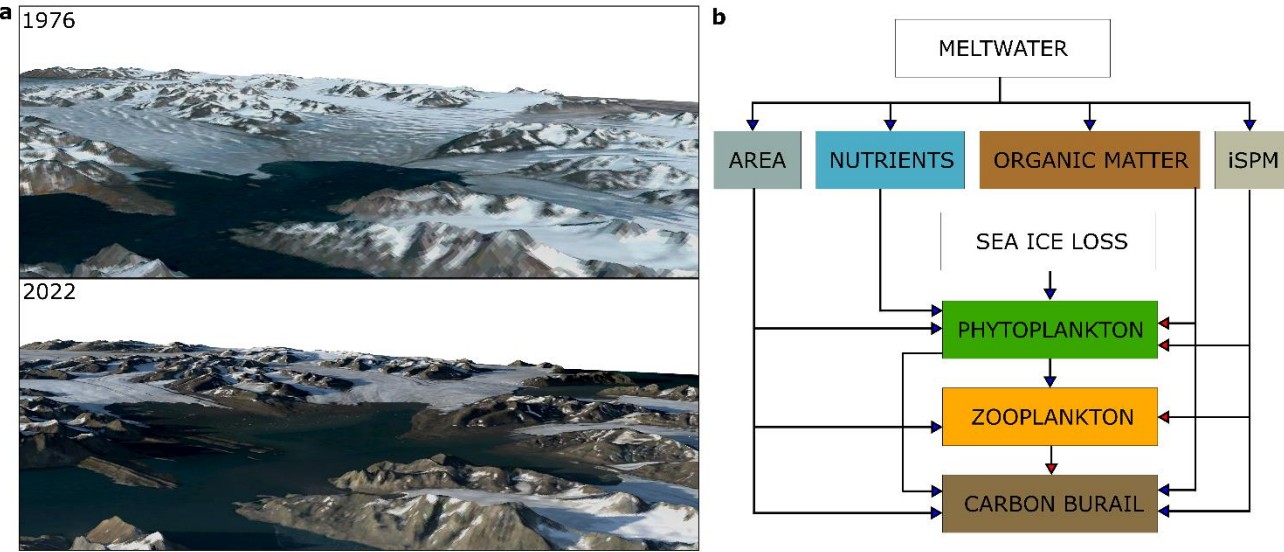


**Figure 7: a) 3D representation of the inner Hornsund bay (Brepollen) in the summer of 1976 and 2022. Landsat satellite images (18/07/1976 and 15/08/2022) were downloaded from https://glovis.usgs.gov/app. Digital elevation model data were downloaded from https://arcticdem.apps.pgc.umn.edu/. b) Schematic representation of the positive (blue arrows) and negative (red arrows) feedback mechanisms influencing biological production and carbon burial in the Arctic fjords.**





**Table 1 Sources of the input data and modelling setup**

| Variable | Data source | Data point | Glacial bay | Coordinates (°N,°E) | Depth (m) |
|---|---|---|---|---|---|
| **Temperature and salinity** | numerical model of Hornsund (HRM) (Jakacki et al., 2017) | H1_08 | Hansbukta | 77.009, 15.624 | 37.87 |
| | | HH1 | | 77.012, 15.624 | 42.45 |
| | | BuP1_05 | Vestre | 77.067, 15.834 | 97.29 |
| | | HA2 | Burgerbukta | 77.074, 15.825 | 61.79 |
| | | HA3 | | 77.079, 15.811 | 61.06 |
| | | HA0 | Austre | 77.082, 15.981 | 57.71 |
| | | HA1 | Burgerbukta | 77.087, 15.967 | 76.93 |
| | | BrS1_02 | Brepollen | 77.029, 16.431 | 55.13 |
| | | BrS1_03 | | 77.048, 16.409 | 47.35 |
| | | HB2 | | 77.067, 16.382 | 44.62 |
| | | H3 | Brepollen | 77.018, 16.503 | 88.02 |
| | | BrH1_03 | | 77.031, 16.528 | 76.32 |
| | | BrH1_04 | | 77.040, 16.581 | 55.45 |
| | | HB1 | | 77.052, 16.571 | 49.55 |
| | | BrSv1_04 | Telegrafbukta | 77.040, 16.581 | 32.81 |
| | | HM2 | | 76.993, 16.638 | 28.46 |
| | | BrM1_04 | Mendeleevbukta | 76.977, 16.562 | 38.97 |
| | | HM1 | | 76.975, 16.575 | 32.65 |
| | | HS2 | Samarinvågen | 76.930, 16.292 | 103.07 |
| | | HS1 | | 76.921, 16.292 | 98.72 |
| **Sea ice concentration and thickness** | S800 model (Albretsen et al., 2017) | H1_08, HH1 (1-2) | | 77.003, 15.637 | |
| | | BuP1_05, HA2, HA3, HA0, HA1 (3-7) | | 77.037, 16.022 | |
| | | BrS1_02, BrS1_03, HB2, H3, BrH1_03, BrH1_04, HB1, BrSv1_04, HM2, BrM1_04, HM1 (8-18) | | 76.993, 16.369 | |
| | | HS1, HS2 (19-20) | | 76.965, 16.239 | |
| **Meteorological data** | Polish Polar Station Hornsund | PPS | | 77.000, 15.550 | |
| **BGC tracers** | mean values from the literature | | | | |
| **Modelling setup** | | | | | |
| **Model** | 1D GOTM-ECOSMO-E2E-Polar | | **Simulation time** | 01/01/2005 – 31/12/2009 | |
| **Spin up** | 5 years (2005-2009 average) | | **Time step** | 30 min | |
| **Depth layers** | 20 | | **Output** | daily average | |





**Table 2 List of parameters, corresponding description, and units used in the model.**

| Abbreviation | Definition | Value | Units |
|---|---|---|---|
| $\tau_{crit}$ | Critical bottom shear stress | 0.07 | $N \cdot m^{-2}$ |
| $\lambda_{d2s}$ | Sedimentation rate if $\tau < \tau_{crit}$ | 3.5 | $m \cdot day^{-1}$ |
| $\lambda_{s2d}$ | Resuspension rate if $\tau \geq \tau_{crit}$ | 26 | $day^{-1}$ |
| $w_D$ | Inorganic SPM sinking rate | 0.8 | $m \cdot day^{-1}$ |
| $k_w$ | Water extinction coefficient | 0.05 | $m^{-1}$ |
| $k_{Chl}$ | Chlorophyll $a$ extinction coefficient | 0.2 | $m^2(m \cdot molC)^{-1}$ |
| $k_{iSPM}$ | Inorganic SPM light extinction coefficient | 0.065 | $m^2g^{-1}$ |
| $k_{DOM}$ | Dissolved organic matter light extinction coefficient | 0.29 | $m^2(m \cdot molC)^{-1}$ |
| $a$ | Photosynthesis efficiency parameter | 0.04 | $(W \cdot m^{-2})^{-1}$ |
| $m_{MB}$ | Macrobenthos mortality rate | 0.03 | $day^{-1}$ |
| $\delta_{bur}$ | Burial rate | 0.0 | $day^{-1}$ |
| $\eta_{bur}$ | Burial efficiency | 0.7 | - |