# Peer review of "The estimates of carbon sequestration potential in an expanding Arctic fjord affected by dark plumes of glacial meltwater (Hornsund, Svalbard)"

_Biogeosciences, 2023_

## Author Comment (AC1)

[Figure]

Sup. Fig. The differences in daily average air temperature (AT), and daily precipitation between the inner and outer fjord based on ERA interim reanalysis (Dee et al., 2011).

Table 1 Variables used to create the assessment and parametrization datasets, and for iSPM simulations

| iSPM reconstruction | iSPM in simulation results | iSPM input to the model | Parametrization of iSPM input |
|---|---|---|---|
| SPM measurements annual sum of PDD | $dC_{iSPM}/dt = C_{iSPM} + C_{iSPMinput}$

general system dynamics | Parametrization 6accPDD (Polish Polar Station) Salinity (from Jakacki et al., 2018) | Sediment Flux measurements 6accPDD (Polish Polar Station) Salinity measurements |

[Figure]

Sup. Fig. Comparison of pelagic primary production from the modeling (station H3, black - iSPM scenario, green - noSPM scenario; average for 2005-2009 with standard deviation) and field studies as reported by Hodal et al., 2012 (Kongsfjorden), Iversen and Seuthe 2011 (Kongsfjorden), Piwosz et al., 2009 (Kongsfjorden and Hornsund), Eilertsen et al., 1989 (Kongsfjorden, Isfjorden, Van Mijenfjorden and Hornsund).

---

## Author Response (AR1)

**Reply to comments of Anonymous Referee #1**

**Dear Reviewer,**

**We appreciate the time and effort that you dedicated to providing feedback on our manuscript and are grateful for the insightful suggestions. Please see below for a point-by-point response to your comments and concerns.**

The modelling study by Szeligowska et al. is an interesting study to implement iSPM into a biogeochemical-hydrodynamic carbon burial model for Arctic fjords. The authors estimate that iSPM is crucial for modelling primary and secondary production in an Arctic fjord, leading to a 50% reduction in production and reduced carbon burial. With increased runoff and glacial retreat iSPM onputs will become increasingly important with climate change, as discussed by the authors. Their overall conclusion of iSPM as a crucial compartment to modelling Arctic fjord systems is supported by a model based on in situ, metereological, and remote sensing data. The study is an important contribution showing the importance of iSPM in modelling carbon cycling in Arctic fjords.

The writing is overall clear, with a few minor grammatical issues. My main concerns lie in the model validation and discussion of the model limitations.

**RESPONSE: We thank the Reviewer for commenting on the value of our manuscript. We improved on the model validation and revised the Discussion according to the suggestions (see details below).**

Major concerns:

The authors indicate a linear change of glacial discharge with climate change (e.g. L49, L374). However, this is only the case in the short term. At some point glacial discharge will decrease. On Svalbard this point seems to have been reached already (See Nowak et al., 2021 https://doi.org/10.5281/zenodo.4294063, Fig. 8).

**RESPONSE: To address this comment we added details in the Discussion about the differences in discharge development as follows (Page 12, L347-349):**

**"While the melting potential is rising, the annual runoff in Svalbard is expected to increase till 2060, then it will likely decrease towards 2100 due to the reduction in glacier storage (Bliss et al., 2014; Van Pelt et al., 2021; Nowak et al., 2021)."**

**Bliss, A., Hock, R., & Radić, V. (2014). Global response of glacier runoff to twenty-first century climate change. Journal of Geophysical Research: Earth Surface, 119(4), 717-730.**

**Van Pelt, W. J., Schuler, T. V., Pohjola, V. A., & Pettersson, R. (2021). Accelerating future mass loss of Svalbard glaciers from a multi-model ensemble. Journal of Glaciology, 67(263), 485-499.**

Air temperature is a key driver for the model, yet, the data are only based on a meteorological station in the outer fjord (e.g. L109), which is quite far away from the inner bay stations. With increasing distance to the west coast I would expect a colder climate. If the authors could show a few comparisons of air temperatures in the inner and outer fjord this argument could hold, otherwise this is a major weakness that should be discussed and ideally quantified (How would colder air temperatures in the inner fjord affect the model?). Also precipitation measurements are likely different in the inner fjords and at higher altitudes and need some critical evaluation of the bias this may introduce (L114).

**RESPONSE: We agree with the reviewer that meteorological forcing is a key driver and that spatial variations could affect the results. In our study, we used the data from the Polish Polar Station located in the outer part of Hornsund (77.00°N, 15.54°E) as a forcing for all the modelling stations, since there is no other long-term monitoring in the fjord (the distribution of meteorological stations in Svalbard is presented in Dahlke et al., 2020). However, the measurements of air temperature in different parts and at different altitudes in Hornsund were performed in 2014/2015 by Araźny et al. (2018). In summer, the inner fjord was colder by 0.6-1°C than values reported for the Polish Polar Station, and the highest difference was observed during winter (around 2°C). It was a result of the sea ice presence in the inner fjord and no heat transfer from open water into land. We also investigated the differences between the inner and outer fjord based on the atmospheric fields derived from ERA-interim reanalysis (Dee et al., 2011). The plots of the differences in daily average air temperature (AT), and daily precipitation between the inner and outer parts of the fjord were added to the Supplementary Material (Supp. Fig. 7). The differences were relatively low, and thus we chose the available in situ measurements from the Polish Polar Station as the more accurate forcing, since the atmospheric models inherently introduce some uncertainty.**

**The points raised here were introduced into the Discussion as follows (Page 15, L448-457):**

"In this study, we used the meteorological forcing from observations performed at the Polish Polar Station located in the outer part of Hornsund for all the modelled stations, since there is no long-term weather monitoring in the inner fjord. A previous study showed that in summer, the air temperature in the inner fjord was lower by 0.6-1°C than values reported for Polish Polar Station, and the highest difference was observed during winter (around 2°C) (Araźny et al., 2018). While the proper atmospheric representation is crucial and, in general, the spatial variations could affect the results, the differences in daily temperatures (AT), and precipitation were relatively low between the inner and outer fjord according to atmospheric fields derived from ERA-interim reanalysis (Dee et al., 2011) (Sup. Fig. 7). It could be related to the fact that Hornsund is a small fjord with an opening mostly influenced by Sørkapp Current transporting Arctic Water from the Barents Sea. The polar front that exists there reduces the advection of warm Atlantic Water into Hornsund. Thus, the entire area retains the arctic properties unlike other West Spitsbergen fjords (Promińska et al., 2017; Cisek et al., 2017)."

Araźny, A., Przybylak, R., Wyszyński, P., Wawrzyniak, T., Nawrot, A. and Budzik, T., 2018. Spatial variations in air temperature and humidity over Hornsund fjord (Spitsbergen) from 1 July 2014 to 30 June 2015. Geografiska Annaler: Series A, Physical Geography, 100(1), pp.27-43.

Cisek, M., Makuch, P. and Petelski, T., 2017. Comparison of meteorological conditions in Svalbard fjords: Hornsund and Kongsfjorden. Oceanologia, 59(4), pp.413-421.

Dahlke, S., Hughes, N.E., Wagner, P.M., Gerland, S., Wawrzyniak, T., Ivanov, B. and Maturilli, M., 2020. The observed recent surface air temperature development across Svalbard and concurring footprints in local sea ice cover. International Journal of Climatology, 40(12), pp.5246-5265.

Dee, D.P., Uppala, S.M., Simmons, A.J., Berrisford, P., Poli, P., Kobayashi, S., Andrae, U., Balmaseda, M.A., Balsamo, G., Bauer, D.P. and Bechtold, P., 2011. The ERA-Interim reanalysis: Configuration and performance of the data assimilation system. Quarterly Journal of the royal meteorological society, 137(656), pp.553-597.

Promińska, A., Cisek, M. and Walczowski, W., 2017. Kongsfjorden and Hornsund hydrography–comparative study based on a multiyear survey in fjords of west Spitsbergen. Oceanologia, 59(4), pp.397-412.

The validation and parametrization data are not independent (ie. Salinity is used as a proxy for iSPM, L177) as also outlined by Wouter van der Niet (Reviewer 1). In general I agree that the model validations need to be clearer and the limitations need to be discussed more critically (e.g. only 5 data points for the regression). Also the comparison of model results in different years than environmental data seem not very robust and need some statistical evaluation and more critical discussion of the limitations (e.g. quantification of interannual variability might be helpful).

**RESPONSE: Model validation is only possible for the results of simulations that correspond to the same time and location as in situ measurements. As it is common for polar research such data were not available, and thus we only performed model assessment as it was described in Methods (2.4 Model assessment) and in Discussion (first paragraph of 4.5 Current limitations and future perspectives).**

**The assessment dataset was based on the correlation between in situ iSPM available for 6 years and the annual sum of daily air temperatures above 0°C (annual PDD AT). We are aware of the limitations of extrapolating this relationship, in particular toward the beginning of the measurements (Fig. 3, iSPM). Thus, the results of correlation for both summer-time and the annual sum of daily air temperatures above 0°C were critically evaluated. The annual sum was chosen as more robust and calculated iSPM concentration was referred to as coarse reconstruction, which limitations were explicitly stated and discussed:**

- **Page 10, L281-288: "The 6-year monitoring dataset of summertime SPM concentration in Hansbukta (Fig. 1c) was not sufficient to show long-term trends. However, average integrated iSPM levels were correlated with both the annual sum of PDD AT (y = 0.061x - 19.549, $R^2$ = 0.68, p<0.05 t-test) and the summertime sum of PDD AT (June to August) (y = 0.221x - 75.047, $R^2$ = 0.78, p<0.05 t-test). Even though the correlation was stronger for the summertime PDD AT, the estimates displayed numerous negative values. However, the annual sum of PDD AT allowed a coarse reconstruction of past conditions and revealed significant increases in iSPM concentration (3.7 g·m$^{-3}$·decade$^{-1}$ in 1979-2022, p<0.001 mMK test). Importantly, within the modelled time range (2005-2009, Fig. 3, grey shade), both iSPM estimates gave similar results in 2006 and 2009 (8.6 and 12.0; 8.1 and 9.8 g·m$^{-3}$, respectively)."**

- **Page 15, L441-444: "While it should be considered that this reconstruction was based on a few years of measurements, which might limit its robustness, particularly towards the beginning of AT measurements, the correlation with annual PDD AT seems to yield reasonable estimates."**

In fact, the 6 data points used for the reconstruction of iSPM carry robust information as they are averages calculated from over a thousand data points distributed over 6 years, 3 summer months, 5 stations and up to 9 depths over the water column, and such complete datasets are rather rare in the Arctic. Both the interannual and seasonal variability for in situ data and simulations were presented e.g. in Fig. 3, 4, 6 and Sup. Fig. 3 and discussed throughout the ms.

Furthermore, the iSPM concentration in the simulation that was later compared with the iSPM from reconstruction is a result of the current iSPM concentration in the model, iSPM input ($dC_{iSPM}/dt = C_{iSPM} + C_{iSPMinput}$), and general dynamics of the model. The iSPM input to the model was created based on the 6-day sum of daily air temperatures above 0°C, salinity (from Jakacki et al., 2018), and parametrization based on inorganic sediment flux and salinity measurements. As illustrated in the table attached, to create the assessment and parametrization datasets we used different variables, thus ensuring their independence.

| iSPM reconstruction | iSPM in simulation results | iSPM input to the model | Parametrization of iSPM input |
|---|---|---|---|
| SPM measurements

annual sum of PDD | $dC_{iSPM}/dt = C_{iSPM} + C_{iSPMinput}$

general system dynamics | Parametrization

6accPDD (Polish Polar Station)

Salinity (from Jakacki et al., 2018) | Sediment Flux measurements

6accPDD (Polish Polar Station)

Salinity measurements |

Moreover, to improve the evaluation of the model we introduced statistical assessment against iSPM measurements in Hansbukta, which was regularly monitored (several times a year) over the whole water column with campaigns covering mostly the main melting season. Due to the robustness of this dataset, we calculated the statistics for data previously presented in Sup. Fig. 3 as requested by another Reviewer. The correlation was added to the Methods (Page 8, L235-239):

"The iSPM concentration at modelled station 2 (HH1) in 2006 and 2009 was also compared with the iSPM field data at monitoring stations M4 (H1_09) and M5 (H1_11) from 2019 (Sup. Fig. 3), which represented environmental conditions (PDD SST, PDD AT, melt season duration, and precipitation in Fig. 3) the closest to the simulation period. Results showed that the model realistically simulated the

seasonal pattern and vertical distribution of the iSPM (rho>0.74, p<0.001 for Spearman's correlation, see Sup. Tab. 4)."

The authors acknowledge subglacial upwelling as an important nutrient source (e.g. L146), yet the 1D model cannot catch the upwelling mechanism and the nutrient inputs by subglacial upwelling. The discussion justifies the lack of nutrient inflow and consumption of nutrients (especially nitrate) with low concentrations in rivers (L486), but most runoff comes from tidewater glaciers which do introduce a lot of nutrients. In other field (e.g. Meire et al., 2023) and modelling studies (Møller et al., 2023) it has been shown that this nutrient source can lead to a significant increase in primary and secondary productivity up to 10s to 100s of km from the glacier front. It seems to be a major limitation of the model that needs more in depth discussions and an estimate of how the model outputs would change if it were included (e.g. higher production without iSPM? Or would more light with less iSPM lead to a quicker nutrient depletion?).

**RESPONSE: While the nutrient input is important, we decided not to include it in the manuscript due to the lack of data for parameterization. Hence, the focus of this study is solely on the iSPM as previously described in Methods (Page 8, L219-221):**

**"We do not provide nutrient input with meltwater due to the lack of data for parametrisation and to disentangle it from the effect of iSPM discharge; thus, the burial rate in the carbon and nitrogen sediment pool (sedCN, Eq. 10) and Si (Eq. 11) is set to 0 to prevent decreasing nutrient concentrations over the simulation time."**

**To address this comment, the Discussion of the limitations was improved as follows (Page 16, L491-493 and Page 16/17, L496-498):**

**"However, Svalbard fjords are relatively shallow, and thus the upwelling pump might not be as efficient as for Greenland fjords or the shallower, nutrient-deficient waters might be transported (Hopwood et al., 2018)."**

**"Even though nutrient input was not provided, setting the nutrient burial rate to 0 allowed keeping the nutrients in the system that would otherwise be excluded, and it could compensate to some degree for the lack of nutrient input with meltwater."**

**The model runs indicating the higher production without iSPM were presented as control runs (noSPM scenario) and they indicate the 50% higher plankton production in comparison to SPM scenario (Fig. 5). Also, more light and  lower**

**iSPM leads to a quicker nutrient depletion which was illustrated for silicate (Fig. 8).**

**Hopwood, M.J., Carroll, D., Browning, T.J., Meire, L., Mortensen, J., Krisch, S. and Achterberg, E.P., 2018. Non-linear response of summertime marine productivity to increased meltwater discharge around Greenland. Nature Communications, 9(1), p.3256.**

The salinity:iSPM relationship is likely different for different catchments and for land- vs marine- terminating glaciers. This should be discussed a bit more (e.g. l178).

**RESPONSE: In this study, we only focus on the marine-terminating glaciers, as only their retreat can result in an expansion of marine habitat, and thus the simulations were not applied to land-terminating glaciers. Nevertheless, as a result of this suggestion, we added the following details to the discussion of the possible differences in the salinity:iSPM relationship (Page 13, L372-373):**

**"The relationship between melting potential and sediment input might differ between catchments, and particularly it could change after glaciers retreat onto land."**

Specific comments:

L32: I am not sure "enhanced underwater light" fits here unless glaciers refers to ice shelves and ice tongues, which would not fit for Hornsund. With new habitats forming after glacier retreat a completely new area opens, which is likely a lot more important than light which is not just enhanced but changes from absent to present once the thick glacial ice covers disappears.

**RESPONSE: The phrase was removed and the sentence was changed as follows (Page 2, L31-33):**

**"Within these newly formed coastal ecosystems, CO2 drawdown by phytoplankton and ice algae is supported by nutrient input from land intensifying the cascade from carbon capture into storage and burial in sediments (Ardyna and Arrigo, 2020; Arrigo et al., 2008; Wadham et al., 2019)."**

L91: How does this standard deviation compare to changes over years? Is the seasonal variability higher than interannual variability?

**RESPONSE: The analysis of long-term changes (46 years) in marine habitat area in Hornsund showed linear increases of around 21.4 km² per decade. Thus, the**

manual digitization error (here 0.23 km$^2$) constitutes only around 1% of decadal changes. The seasonal variability was not taken into consideration in this study and the analysis was narrowed to summer months.

L95: How do you estimate the depth at the new habitats (After 2010) which is a key to this study? Did you assume the bottom depth at the glacier front in 2010 is the same in the new area once the glacier retreats? Is there any support for this assumption, or should it be mentioned as study limitation/uncertainty?

**RESPONSE: The simulations were run for 2005-2009 and we did not estimate the depth after 2010. Thus, the changes in volume in Fig. 3 are provided only till 2010.**

**It was clarified as follows (Page 3, L91-94):**

**"Marine habitat volume was calculated based on digitized area and bathymetry data from Hornsund (grid size 100 m) (Moskalik et al., 2014) using the zonal statistics method in ArcGIS Pro 2.8.0. Marine habitat volume was calculated until 2010, since the bathymetric data were not available for the glacial bays that emerged after 2010."**

L125: How was the iSPM concentration measured? Based on water samples taken with Niskin bottles? At what depths?

**RESPONSE: The water samples were collected using a Hydrobios Free Flow 1L Niskin Bottle at several water depths as described in Moskalik et al. (2018). The depths were distributed over the water column and depended on the maximum depth at the stations as provided in the original dataset (https://dataportal.igf.edu.pl/dataset/spm-hornsund). This dataset was submitted to PANGAEA data publisher and a data paper is in preparation and will be cited here if the timing allows.**

**To address this comment, we changed the description of Methods as follows (Page 4, L120-122):**

**"Datasets for suspended particulate matter (SPM), sediment flux, and salinity collected in Hansbukta (2015-2021) at long-term monitoring stations (Fig. 1c, Sup. Tab. 2) were downloaded from https://dataportal.igf.edu.pl/group/longhorn (see detailed description in Moskalik et al., 2018)."**

L98: This sentence is confusing. Which dataset is used exactly? Currently it sounds like "a temperature dataset delivered temperature data".

**RESPONSE: Arctic Sea and Ice Surface Temperature is the name of the dataset, and it was cited as requested by the authors. To avoid confusion we changed this sentence as follows (Page 4, L96-97):**

**"Sea and ice surface temperature (SST) and sea-ice concentration (SIC) variables were extracted from the Arctic Sea and Ice Surface Temperature dataset (L4, 5km, daily)."**

L107: Wouldn't submarine melt be mostly affected by deeper water layers? Especially in winter, submarine melt can still be substantial in some fjords even though the SST is below 0C. If you can show CTD profiles or refer to a study with CTD profiles that show that SST is a good proxy for the overall heat at the glacier-seawater interface this argument could hold, otherwise I am skeptical.

**RESPONSE: The glacial bays in Svalbard are typically relatively shallow (up to 100 m depth) and the water column is well-mixed in winter, thus temperature is relatively stable within the whole water column. The same proxy was used by Blaszczyk et al. (2021, 2023) for analysis of glacier recession in Hornsund where a comparison of SST with mean temperature from CTD profiles was shown. The CTD profiles collected in Hornsund are available at https://dataportal.igf.edu.pl/dataset/temperature-salinity-turbidity-oxygen-depth-hornsund. The data on surface and subsurface sea temperatures for other glacial bays in Svalbard were illustrated in Luckman et al. (2015).**

**The references were added in the text as follows (Page 4, L104-106):**

**"The sum of all daily SST>0°C (positive degree days, PDD SST) was calculated for each year (annual) and each melt season (summertime, June-August) as a proxy for submarine melt potential (Luckman et al., 2015; Błaszczyk et al., 2021, 2023)."**

**Błaszczyk, M., Jania, J.A., Ciepły, M., Grabiec, M., Ignatiuk, D., Kolondra, L., Kruss, A., Luks, B., Moskalik, M., Pastusiak, T. and Strzelewicz, A., 2021. Factors controlling terminus position of Hansbreen, a tidewater glacier in Svalbard. Journal of Geophysical Research: Earth Surface, 126(2), p.e2020JF005763.**

**Błaszczyk, M., Moskalik, M., Grabiec, M., Jania, J., Walczowski, W., Wawrzyniak, T., Strzelewicz, A., Malnes, E., Lauknes, T.R. and Pfeffer, W.T., 2023. The Response of Tidewater Glacier Termini Positions in Hornsund (Svalbard) to Climate Forcing, 1992-2020. Journal of Geophysical Research: Earth Surface, p.e2022JF006911.**

**Luckman, A., Benn, D.I., Cottier, F., Bevan, S., Nilsen, F. and Inall, M., 2015. Calving rates at tidewater glaciers vary strongly with ocean temperature. Nature communications, 6(1), p.8566.**

L145: I agree that advection from outside is limited, but advection from the glacial (subglacial water) should play a role. Please differentiate.

**RESPONSE: It was changed to (Page 5, L144-146):**

**"However, advection of Atlantic Water is considered to be limited in Hornsund in comparison to other West Spitsbergen fjords, particularly in the inner bays, whereas upwelling is most important up to 500 m of distance from the glacier fronts (Pasculli et al., 2020)."**

L151: Does the 3D hydrodynamic model differentiate between surface and subglacial meltwater discharge?

**RESPONSE: The 3D hydrodynamic model for Hornsund does not. Such a model was currently only implemented in Hansbukta in 2D (De Andrés et al., 2018; 2021), and the 3D developments are ongoing, with one study under review (https://tc.copernicus.org/preprints/tc-2023-144/).**

L235: Why was the model period set to 2005-2009 when there is a lack of data? It seems that the other data (e.g. meteorological data) are available until 2018. Also Senintel 3 data in a higher resolution would be available since 2016. This needs clarification.

**RESPONSE: The simulation time is primarily limited by the input data from the hydrodynamic model (i.e. temperature and salinity) as was previously described in Discussion (Page 16, Line 471-475):**

**"Even though Hornsund is amongst the most studied Svalbard fjords, our study was limited to 5 years period due to the lack of long-term input data for temperature and salinity (Jakacki et al., 2017; Torsvik et al., 2019) as most of the hydrodynamic models do not simulate coastal zones with sufficient horizontal resolution and they do not consider changes in the extent of glacial bays. It should also be considered that sea ice concentration and thickness were extracted from the closest data points available, and thus sea ice conditions might have been different in the glacial bays."**

**All the hydrodynamic models available for West Spitsbergen fjords were run for 2005-2010 (e.g. Jakacki et al., 2017; Sundfjord et al., 2017; Torsvik et al., 2019) due**

to the lack of forcing data for other years. While it is an inconvenience here, we believe that further research will address this issue in the future.

Satellite data products available at Copernicus were verified during preparation of this manuscript, and they do not represent the SPM and chlorophyll *a* concentrations properly in the fjords or there are data gaps due to clouds. The inner fjords are optically complex and the algorithms used to create these products were not calibrated for the glacial bays. On the other hand, calibration based on raw satellite data and in situ measurements could be performed, but is beyond the scope of this study and was not yet done for Hornsund. Satellite data also do not represent subsurface peaks of SPM that are present in Hornsund. Due to these reasons, the reconstruction of iSPM based on long-term monitoring was assumed as the most accurate for model assessment as described in Discussion (Page 15, Line 436-447):

"While the coupled physical-biogeochemical model with newly implemented iSPM input performed well according to our assessment, the field data for model parametrisation and validation were not available for the simulated period (2005–2009), whereas remotely-sensed products for iSPM concentration did not cover the inner fjords and were frequently limited by clouds. Despite that, the reconstructions of previous conditions and assessment based on the two complementary datasets collected in recent years (2015–2021, SPM and sediment flux) suggest that the simulated spatial and temporal dynamics of both inorganic and organic SPM was rather realistic and in line with the current knowledge of the West Spitsbergen marine ecosystem. While it should be considered that this reconstruction was based on a few years of measurements, which might limit its robustness, particularly towards the beginning of AT measurements, the correlation with annual PDD AT seems to yield reasonable estimates. A recent multi-year study (2010-2016) in another West Spitsbergen fjord (Kongsfjorden) also indicated the relationship between particle fluxes and air temperature above the melting point (D'Angelo et al., 2018). Notably, the satellite data products calibrated for the glacial bays should become available (Klein et al., 2021; Walch et al., 2022) to verify the long-term trends in the iSPM discharge revealed in this study."

L336: Why is silicate limitation considered, but not nitrate limitation? The model has flagellates as a separate phytoplankton group, which is not dependent on silicate.

RESPONSE: Both the nitrogen and phosphorus limitations were also taken into account in the model. They were not presented in Fig. 8, since the figure was already complex and the silicate limitation is an important indicator of the

spring-summer bloom transition. However, the nitrogen and phosphorus limitations were now added to the plot and the Methods were adjusted accordingly (Page 9, L257-261):

"The influence of iSPM discharge on the ecosystem dynamics was exemplified by presenting biomass of ice algae (IA) and macrobenthos (MB), as well as biomass of phytoplankton (PHY), zooplankton (ZOO), silicate, phosphorus, nitrogen and light limitation index (SIL, PLI, NLI, LLI) integrated for the whole water column at three modelled stations (2, 9, 14) that were comparable due to similar depths (42.45 – 49.55 m), but presented low, intermediate and high level of summertime iSPM input."

L342: Hornsund is also a West Spitsbergen fjord. Thus, I suggest writing: .. in "other" West Spitsbergen fjords (...

RESPONSE: The studies referenced were also performed in Hornsund, thus we will not introduce this change.

L351f: I don't see how the findings of increasing PDD and melt season length in Hornsund can be important for predictions in other regions with higher temperatures? Does it mean that the finding of higher temperatures (=increased PDD and melting season length) leading to melting should be applied to other systems? This needs clarification.

RESPONSE: This sentence was meant to highlight that the findings from this study related to ecosystem functioning could be important for other regions experiencing similar changes in the environmental conditions. It was now moved to the Conclusions (Page 17, L515-517).

L361f: This statement needs a reference.

RESPONSE: The loss of the ice bridge between Brepollen and Hambergbukta will change the hydrographic situation in the South Spitsbergen and Hornsund region due to the formation of a new island and a strait. One of the scenarios was described by Grabiec et al., (2017) and it is mostly related to strong tidal currents in the newly formed strait due to differences in sea water level. In our opinion, there are at least two other possible scenarios. First, cold water from Storfjorden could be transported into Hornsundbanken and the West Spitsbergen coast through the newly created "Horsnund Strait". Consequently, Sorkapland could become even more isolated from warm Atlantic Water influence than currently. The second scenario is the opposite, i.e. there could be a clock-wise circulation of water masses around the new "Sorkapland Island". In such a situation warm

**Atlantic Water would be transported through the new "Hornsund Strait". These two scenarios are only hypothetical and were not described before in any article.**

**Grabiec, M., Ignatiuk, D., Jania, J.A., Moskalik, M., Głowacki, P., Błaszczyk, M., Budzik, T. and Walczowski, W., 2017. Coast formation in an Arctic area due to glacier surge and retreat: The Hornbreen–Hambergbreen case from Spistbergen. Earth Surface Processes and Landforms, 43(2), pp.387-400.**

L368: This iSPM increase mentioned based on in situ data right? I suggest clarifying it.

**RESPONSE: It was clarified as follows (Page 12, L363-364):**

**"In this study, reconstructed iSPM concentration increased after 2013 and further rises are expected (3.7 g·m$^{-3}$·decade$^{-1}$ integrated for the water column in summer)."**

L395f: I suggest going into more details in the comparison of the model outputs with field studies. How does it compare to turbid vs clear fjord systems?

**RESPONSE: A plot comparing the model outputs of pelagic primary production with the field studies was added to the Supplementary Material (Sup. Fig. 6). The primary production and standing stocks of protists and zooplankton are typically reduced in the glacial bays in summer in comparison to clear waters, as also presented in this study.**

L398f: I don't agree with this argument. Phytoplankton blooms start when sea ice breaks up, by then sea ice algae are simply substrate limited. When sea ice is stable Phytoplankton usually does not compete for nutrients under the ice because they have too little light. If the authors disagree with me I would like to see a reference to a study that shows phytoplankton competing for nutrients with sea ice algae.

**RESPONSE: We removed the mentioned part of the sentence as follows (Page 13, L394-395):**

**"Ice algae did not seem to be negatively affected by iSPM and, as modeling results suggest, their biomass was slightly higher in the SPM scenario than in the noSPM scenario".**

L406: What is this expectation based on?

**RESPONSE: It is based on general rules of the flow of energy and matter through the trophic levels, with app. only 10% transfer efficiency form one trophic level to the next.**

L416: This is too speculative. If you can find a supporting reference it might still be ok to include, but I dont see that your study shows this decreased food web complexity.

**RESPONSE: The references were provided (Page 14, L412-414):**

**"Thus, we speculate that sediment discharge to polar coastal zones might result in less complex food webs, constituted by species better adapted to high iSPM concentrations and sedimentation rates as shown for Antarctic benthos (Clark et al., 2013; Krzemińska and Kukliński, 2018; Sahade et al., 2015)."**

**Clark, G.F., Stark, J.S., Johnston, E.L., Runcie, J.W., Goldsworthy, P.M., Raymond, B. and Riddle, M.J., 2013. Light-driven tipping points in polar ecosystems. Global Change Biology, 19(12), pp.3749-3761.**

**Krzeminska, M. and Kuklinski, P., 2018. Biodiversity patterns of rock encrusting fauna from the shallow sublittoral of the Admiralty Bay. Marine environmental research, 139, pp.169-181.**

**Sahade, R., Lagger, C., Torre, L., Momo, F., Monien, P., Schloss, I., Barnes, D.K., Servetto, N., Tarantelli, S., Tatián, M. and Zamboni, N., 2015. Climate change and glacier retreat drive shifts in an Antarctic benthic ecosystem. Science Advances, 1(10), p.e1500050.**

L432: What are these anticipated negative effects? So far we mostly see an increase in NPP with decreasing sea ice.

**RESPONSE: Here, the 'ice loss' is referred to glaciers,not sea ice. The negative effects such as the increasing SPM discharge or loss of upwelling were discussed throughout the manuscript.**

**The sentence was clarified as follows (Page 15, L427-429):**

**"Due to the anticipated negative effects of glacier ice loss (Hunter, 2022), here we show that part of the potential gains in carbon sequestration related to the newly ice-free areas..."**

L435: here might be a good place to mention the limitaitons of the model in more detail (ie. Modelling of nutrients).

**RESPONSE: The model limitations were discussed in detail in the next section of the Discussion (4.5 Current limitations and future perspectives).**

L488: I do not agree. You mention that nutrients in meltwater and rivers are low, but then also that most glaciers in Hornsund are marine terminating, where subglacial upwelling is a key nutrient source increasing NPP substantially. Also the iSPM to Sal relationship would be very different in a marine terminating vs land terminating systems.

**RESPONSE: In this study, we only focused on the marine-terminating glaciers as only their retreat can result in expansion of marine habitat. Even though the simulations were not applied to land-terminating glaciers, we added a sentence related to changing iSPM to salinity relationship (Page 13, L372-373):**

**"The relationship between melting potential and sediment input might differ between catchments, and particularly it could change after glaciers retreat onto land."**

**While the nutrient upwelling is important in Greenland, it might be of less importance in Svalbard. It was discussed as follows (Page 16, L490-492; Page 17, L495-497):**

**"However, Svalbard fjords are relatively shallow, and thus the upwelling pump might not be as efficient as for Greenland fjords or the shallower, nutrient deficient waters might be transported (Hopwood et al., 2018)."**

**"Even though nutrient input was not provided per se, setting nutrient burial rate to 0 allowed keeping the nutrients in the system that would otherwise be excluded and it could to some degree compensate for lack of nutrient input with meltwater."**

**Taking all that into account, the overall bias introduced by not providing nutrient input in our simulations might be relatively low.**

**Hopwood, M.J., Carroll, D., Browning, T.J., Meire, L., Mortensen, J., Krisch, S. and Achterberg, E.P., 2018. Non-linear response of summertime marine productivity to increased meltwater discharge around Greenland. Nature Communications, 9(1), p.3256.**

Grammatical suggestions:

Line 17: Formation of "a" new marine habitat OR Formation of new marine "habitats"

L62: I suggest using "best" instead of "most"

L65: with "a" newly implemented iSPM group.

L255: multiplied "with" the average

L289: allowed "a" coarse reconstruction

L354: "where" mass loss cannot be…

L494: "field" measurements

**RESPONSE: The corrections were introduced to the text as suggested.**

**Reply to comments of Wouter van der Niet**

**Dear Wouter van der Niet,**

**We appreciate the time and effort that you dedicated to providing feedback on our manuscript and are grateful for the insightful suggestions. Please see below for a point-by-point response to your comments and concerns.**

*This review was prepared as part of graduate program course work at Wageningen University, and has been produced under supervision of Rúna Magnússon. The review has been posted because of its good quality, and likely usefulness to the authors and editor. This review was not solicited by the journal.*

Szeligowska et al. present an interesting paper about the effect of iSPM concentration on carbon burial fluxes in newly ice free areas in fjords. They use meteorological and satellite data to make an assessment of the melting rate and dynamics in Hornsund. Most importantly a 1D coupled physical biogeochemical model including the iSPM concentration is presented. Model simulations show that the newly ice-free areas contributed significantly to primary production, secondary production and carbon burial. Plankton primary production and secondary production were halved, whereas the carbon burial had decreased with 16%. Based on the results it is rightfully concluded that iSPM concentration is an important factor in modelling carbon fluxes of newly ice free areas in arctic fjords.

Including iSPM concentration in the modelling is an innovation in the science of carbon flux modelling in polar fjords. Previous studies did not include iSPM in carbon flux modelling in polar fjords. Moreover this paper shows that iSPM concentrations are important to include in carbon flux modelling of polar fjords as it decreased the carbon burial with 16% in the model simulations. Another aspect that shows the value of this paper is that it presents a first attempt at modelling the carbon burial flux by taking into account pelagic, sympagic and benthic factors. Therefore the study provides important building blocks for expanding the modelling to a 3D high resolution model, advancing the carbon flux understanding of polar fjords. Considering this advancement, the study fits the scope of the Biogeosciences Journal.

Overall the writing is clear. Specifically, the limitations and assumptions  of the 1D-approach of the model are well explained. For example, the 1D approach does not allow representation of advection and circulation in the fjord. I liked how these limitations were identified and investigated how this affects the results. Also, the effect of not including variable nutrient input in the model was discussed well. There are however a few important aspects of the writing that still need improvement. The main message of the paper and the importance of carbon burial in newly-ice free areas are

weakly conveyed to the reader. I also have some major concerns regarding the validation of the model. I would like to see statistical quantification on the validation of the model. Both issues will need to be addressed sufficiently before the manuscript is accepted.

**RESPONSE: We would like to thank you for commenting on the value of our manuscript. We have improved on the model validation and revised the manuscript according to the suggestions (see details below).**

Major Comments:

1. I am particularly concerned about the model validation. In the validation, reconstructed mean summertime iSPM concentrations are reconstructed based on iSPM concentration of later years and PDD AT. However, the model calculates iSPM concentration based on PDD AT aswell. So this is not independent data. Moreover, this results in only 5 data points of the 5 years. Correlation is found, but by using only 5 data points this is not robust.

**RESPONSE: Model validation is only possible for the results of simulations that correspond to the same time and location as in situ measurements. As it is common for polar research such data were not available, and thus we only performed model assessment as it was described in Methods (2.4 Model assessment) and Discussion (first paragraph of 4.5 Current limitations and future perspectives).**

**The assessment dataset was based on the correlation between in situ iSPM available for 6 years and the annual sum of daily air temperatures above 0°C (annual PDD AT). We are aware of the limitations of extrapolating this relationship, in particular toward the beginning of the measurements (Fig. 3, iSPM). Thus, the results of the correlation for both summer-time and the annual sum of daily air temperatures above 0°C were critically evaluated. The annual sum was chosen as more robust and the calculated iSPM concentration was referred to as coarse reconstruction, which limitations were explicitly stated and discussed:**

- **Page 10, L281-288: "The 6-year monitoring dataset of summertime SPM concentration in Hansbukta (Fig. 1c) was not sufficient to show long-term trends. However, average integrated iSPM levels were correlated with both the annual sum of PDD AT (y = 0.061x - 19.549, $R^2$ = 0.68, p<0.05 t-test) and the summertime sum of PDD AT (June to August) (y = 0.221x - 75.047, $R^2$ = 0.78, p<0.05 t-test). Even though the correlation was stronger for the summertime PDD AT, the estimates displayed numerous negative values.**

> However, the annual sum of PDD AT allowed a coarse reconstruction of past conditions and revealed significant increases in iSPM concentration (3.7 g·m$^{-3}$·decade$^{-1}$ in 1979-2022, p<0.001 mMK test). Importantly, within the modelled time range (2005-2009, Fig. 3, grey shade), both iSPM estimates gave similar results in 2006 and 2009 (8.6 and 12.0; 8.1 and 9.8 g·m$^{-3}$, respectively)."

- **Page 15, L441-444: "While it should be considered that this reconstruction was based on a few years of measurements, which might limit its robustness, particularly towards the beginning of AT measurements, the correlation with annual PDD AT seems to yield reasonable estimates."**

The 6 data points used for the reconstruction of iSPM carry robust information as they are averages calculated from over a thousand data points distributed over 6 years, 3 summer months, 5 stations and up to 9 depths over the water column, and such complete datasets are rather rare in the Arctic. Both the interannual and seasonal variability for in situ data and simulations were presented e.g. in Fig. 3, 4, 6 and Sup. Fig. 3 and discussed throughout the manuscript.

Furthermore, the iSPM concentration in the simulation that was later compared with the iSPM from reconstruction is a result of the current iSPM concentration in the model, iSPM input ($dC_{iSPM}/dt = C_{iSPM} + C_{iSPMinput}$), and general dynamics of the model. The iSPM input to the model was created based on the 6-day sum of daily air temperatures above 0°C, salinity (from Jakacki et al., 2018), and parametrization based on inorganic sediment flux and salinity measurements. As illustrated in the table below, to create the assessment and parametrization datasets we used different variables, thus ensuring their independence.

| iSPM reconstruction | iSPM in simulation results | iSPM input to the model | Parametrization of iSPM input |
|---|---|---|---|
| SPM measurements

annual sum of PDD | $dC_{iSPM}/dt = C_{iSPM} + C_{iSPMinput}$

general system dynamics | Parametrization

6accPDD (Polish Polar Station)

Salinity (from Jakacki et al., 2018) | Sediment Flux measurements

6accPDD (Polish Polar Station)

Salinity measurements |

Moreover, the model output of station 2 in 2006 and 2009 is assessed to be realistic (line 241) when compared to measurements of 2019. This is not convincing because

they are different years and primarily because there is no statistical quantification on the accuracy of this comparison. Lastly, the spatial patterns were found to be in line with the simulation results by comparing the simulations to field measurements of 2017. There is no quantification of the statistics of this comparison either. The result of the poor validation is that it undermines the credibility of the simulation results of spatial patterns in figure 5 and the temporal patterns in figure 6.

In order to make the validation of the model more credible, I would like to see quantification and elaboration on the statistics of the comparison between (1) the model simulation of stations 4 and 5 in 2006 and 2009 and measurements of 2019 and (2) between the model simulations and field measurements of 2017 (spatial variation). More specifically, this includes a table of the R-squared and p-values of the linear regression between the model simulation of stations 4 and 5 in 2006 and 2009 and the measurements of 2019. This should be carried out for each individual date used in figure 2 of the supplement. Secondly, I would like to see at least the R-squared, p-value and coefficient of correlation of the linear regression between the model simulated iSPM concentration in all modelled stations and field measurements of iSPM concentration in 2017.

**RESPONSE: The iSPM in Hansbukta was regularly monitored (several times a year) over the whole water column with campaigns covering mostly the main melting season (Sup. Fig. 3). Thus, even though the data were not collected in the same year as the simulation time, we chose the year of measurements with the most similar meteorological conditions as explained in the Methods. Due to the robustness of this dataset, we calculated the statistics as requested. The correlation was added to the Methods (Page 8, L235-239):**

**"The iSPM concentration at modelled station 2 (HH1) in 2006 and 2009 was also compared with the iSPM field data at monitoring stations M4 (H1_09) and M5 (H1_11) from 2019 (Sup. Fig. 3), which represented environmental conditions (PDD SST, PDD AT, melt season duration, and precipitation in Fig. 3) the closest to the simulation period. Results showed that the model realistically simulated the seasonal pattern and vertical distribution of the iSPM (rho>0.74, p<0.001 for Spearman's correlation, see Sup. Tab. 4)."**

**In 2017, the data for all glacial bays in Hornsund were collected only once in summer and they only covered the surface water layer. Due to the high variability of iSPM concentration throughout the season, such data are not suitable for calculating the correlation.**

1.  The second major issue is that two aspects should be highlighted more in the writing. Both the main message and significance of carbon burial in newly ice-free fjords are weakly conveyed to the reader.

In line 28, 29 and 35 it is stated that the burial of carbon in newly ice-free fjord sediments is an important pathway for carbon sequestration. The importance of this pathway is not quantified however. The result is that the reader is not convinced of the relevance of this paper and its contribution to advancing the understanding of carbon fluxes and climate change. I recommend to quantify the contribution of fjords to carbon sequestration in marine sediments. For example this is done by Bianchi et al. (2020): "over the last 100,000 years, 12% of continental margin sediments have been stored in fjords, and likely have a nearly equal contribution to total marine OC burial". A quantification similar to this would suffice.

**RESPONSE: While we agree with the comment, such quantification is beyond the scope of this study as it requires a global and in-depth analysis of numerous fjord systems that can differ significantly. To our knowledge, the area of most of the glaciated fjords was not updated and carbon burial rates were not commonly measured in the newly formed glacial bays even in well-studied polar fjords (Włodarska-Kowalczuk et al., 2019; Koziorowska et al., 2018), thus reducing the possibility of such generalizations. The unquantified potential of expanding polar fjords is the core of the current study where we aimed to define this knowledge gap.**

I conclude that the main message of the paper is: "iSPM input from glacial meltwater is an important factor in more accurately resolving carbon fluxes. Therefore it should be implemented in current ocean models applied to arctic fjords coastline systems." In the abstract this is not clearly stated. It is only stated that enhanced land-ocean connectivity should be investigated further. I recommend to put the main message explicitly before this line or replace this line with a more clear sentence covering the main message.

**RESPONSE: From our perspective, the most important message from this study would be that the net effects of glacial retreat on productivity and carbon burial in the polar fjords might be positive despite the negative influence of iSPM. Thus, this suggestion was not introduced.**

In the conclusion the main message is there in line 510. However I suggest to change the context a bit because the two sentences before conclude that there is still a lot of uncertainties: "Considerable uncertainties remain, in particular related to the petrogenic organic carbon release". Straight after the main message, the importance of open long-term datasets is stressed (line 511-514). I suggest to move up the sentence

in line 510 to right after "…. the emergence of carbon sinks due to the formation of newly ice-free areas". This way the order is more logical for the reader and the main message is highlighted better.

**RESPONSE: The sentence was transferred as suggested.**

1. Another concern is the meteorological data that was used. Air temperature and precipitation data was used from the Polish polar station Hornsund. This station is located at the extreme west of the Hornsund fjord and at 50 kilometres distance from the eastern-most modelled stations. The meteorological conditions at the west are strongly dominated by the proximity of the relatively warm ocean. Whereas the influence of the sea decreases more inland at the east side of the study area. The meteorological data are thus not representative for the entire study area.

**RESPONSE: We agree with the reviewer that meteorological forcing is a key driver and that spatial variations could affect the results. In our study, we used the data from the Polish Polar Station located in the outer part of Hornsund (77.00°N, 15.54°E) as a forcing for all the modelling stations, since there is no other long-term monitoring in the fjord (the distribution of meteorological stations in Svalbard is presented in Dahlke et al., 2020). However, the measurements of air temperature in different parts and at different altitudes in Hornsund were performed in 2014/2015 by Araźny et al. (2018). In summer, the inner fjord was colder by 0.6-1°C than values reported for Polish Polar Station, and the highest difference was observed during winter (around 2°C). It was a result of the sea ice presence in the inner fjord and no heat transfer from open water into land. We also investigated the differences between the inner and outer fjord based on the atmospheric fields derived from ERA-interim reanalysis (Dee et al., 2011). The plots of the differences in daily average air temperature (AT), and daily precipitation between the inner and outer parts of the fjord were added to the Supplementary Material (Sup. Fig. 7). The differences were relatively low, and thus we chose the available in situ measurements from the Polish Polar Station as the more accurate forcing, since the atmospheric models inherently introduce some uncertainty.**

**The points raised here were introduced into the Discussion as follows (Page 15, L448-457):**

**"In this study, we used the meteorological forcing from observations performed at the Polish Polar Station located in the outer part of Hornsund for all the modelled stations, since there is no long-term weather monitoring in the inner fjord. A previous study showed that in summer, the air temperature in the inner**

fjord was lower by 0.6-1°C than values reported for Polish Polar Station, and the highest difference was observed during winter (around 2°C) (Araźny et al., 2018). While the proper atmospheric representation is crucial and, in general, the spatial variations could affect the results, the differences in daily temperatures (AT), and precipitation were relatively low between the inner and outer fjord according to atmospheric fields derived from ERA-interim reanalysis (Dee et al., 2011) (Sup. Fig. 7). It could be related to the fact that Hornsund is a small fjord with an opening mostly influenced by Sørkapp Current transporting Arctic Water from the Barents Sea. The polar front that exists there reduces the advection of warm Atlantic Water into Hornsund. Thus, the entire area retains the arctic properties unlike other West Spitsbergen fjords (Promińska et al., 2017; Cisek et al., 2017)."

Araźny, A., Przybylak, R., Wyszyński, P., Wawrzyniak, T., Nawrot, A. and Budzik, T., 2018. Spatial variations in air temperature and humidity over Hornsund fjord (Spitsbergen) from 1 July 2014 to 30 June 2015. Geografiska Annaler: Series A, Physical Geography, 100(1), pp.27-43.

Cisek, M., Makuch, P. and Petelski, T., 2017. Comparison of meteorological conditions in Svalbard fjords: Hornsund and Kongsfjorden. Oceanologia, 59(4), pp.413-421.

Dahlke, S., Hughes, N.E., Wagner, P.M., Gerland, S., Wawrzyniak, T., Ivanov, B. and Maturilli, M., 2020. The observed recent surface air temperature development across Svalbard and concurring footprints in local sea ice cover. International Journal of Climatology, 40(12), pp.5246-5265.

Dee, D.P., Uppala, S.M., Simmons, A.J., Berrisford, P., Poli, P., Kobayashi, S., Andrae, U., Balmaseda, M.A., Balsamo, G., Bauer, D.P. and Bechtold, P., 2011. The ERA-Interim reanalysis: Configuration and performance of the data assimilation system. Quarterly Journal of the royal meteorological society, 137(656), pp.553-597.

Promińska, A., Cisek, M. and Walczowski, W., 2017. Kongsfjorden and Hornsund hydrography–comparative study based on a multiyear survey in fjords of west Spitsbergen. Oceanologia, 59(4), pp.397-412.

Considering that the air temperature is one of the two parameters used to derive the iSPM concentration in the model, having representative air temperature data is crucial for the accuracy of the model simulations. Therefore I suggest to include in the discussion chapter a statement of what the meteorological differences are in the east part of Hornsund compared to the polish polar station in the west of Hornsund.

Secondly, the influence of these meteorological differences on the model simulations has to be pointed out clearly.

**RESPONSE: This point was addressed with the previous comment.**

Minor Comments:

1. A fixed carbon to Chlorphyll a ratio was used in the model. In contrast to a dynamic carbon to chlorphyll *a* as is explained in Yumruktepe et al. (2022). Light attenuation by chlorophyll is therefore represented slightly inaccurately in this study. This might be relevant to mention in the discussion chapter.

**RESPONSE: We realized that due to the numerous parallel developments of different components of the ECOSMO II, we did not explicitly state that the version used in this study also included chlorophyll *a* as a prognostic variable allowing a flexible chlorophyll-to-carbon ratio as described in Yumruktepe et al., (2022). It was now corrected as follows (Page 5, 133-137):**

**"ECOSMO-E2E-Polar version represents the three main nutrient cycles (nitrogen, phosphorus, and silica) in the pelagic and sympagic systems, three functional groups of primary producers (ice algae, diatoms, and flagellates), two zooplankton groups (micro- and meso-), one macrobenthos group, and chlorophyll a as a prognostic variable allowing a flexible chlorophyll-to-carbon ratio. The ECOSMO developments are fully described in Benkort et al. (2020), Daewel et al. (2018), Daewel and Schrum (2013), Yumruktepe et al. (2022)."**

1. The carbon burial efficiency of 70% was used, but can be highly variable in different fjords as found by Koziorowska et al., (2018). It can be useful to point this out in the discussion chapter.

**RESPONSE: Thank you for pointing this out. It was introduced in the Methods and Discussion as follows (Page 8, L226-227; Page 15, L432-433):**

**"The carbon burial potential (CB, Eq. 12) was calculated as 70% burial efficiency of the carbon and nitrogen sediment accumulation rate as previously reported for Hornsund (Koziorowska et al., 2018):"**

**"Importantly, the carbon burial efficiency is highly variable and differs between fjords (Koziorowska et al., 2018), thus limiting the direct generalizations."**

1. In figure 7b, the colours (red/blue) in the arrows, indicating positive and negative feedback mechanisms, are hard to distinguish.

**RESPONSE: The arrows were changed for clarity as requested.**

---

## Author Response (AR2)

**Reply to comments of Anonymous Referee #1**

I want to thank the authors for considering and addressing all my comments in detail. The manuscript improved considerably and can be accepted after some last minor revisions.

**Dear Reviewer,**

**Thank you for appreciating the improved manuscript and providing feedback on the new version. Please see below a response to your comments and concerns.**

I appreciate the added discussion about non-linear changes in freshwater runoff on Svalbard. However, the introduction is still only mentioning an increase in freshwater runoff (L49). I suggest specifying that this increase is a short term (until 2060) increase.

**The increasing meltwater runoff is not mentioned in the Introduction. However, the short term increase was already specified in the Discussion as previously requested (L348):**

**"While the melting potential is rising, the annual runoff in Svalbard is expected to increase till 2060, then it will likely decrease towards 2100 due to the reduction in glacier storage (Bliss et al., 2014; Van Pelt et al., 2021; Nowak et al., 2021)."**

The argument that subglacial upwelling is likely not a major nutrient source due to the shallow grounding line depth is acceptable for Hornsund and probably for most Svalbard fjords with shallow tidewater glacier. However, this means that the model is also limited to similar fjords (shallow tidewater glaciers, where subglacial upwelling is not the most important summer nutrient source), which does not include most Greenland fjords with deep tidewater glaciers, or Fjords with land-terminating or deglaciated catchment. Also comparisons with Antarctica are quite speculative since the drivers of primary production are very different (e.g. iron limitation, deep wind mixing). This needs to be clarified throughout the manuscript and generalizations such as in L515ff should be avoided or discussed more carefully. I don't think the study becomes less important by specifying more clearly for which system it is valid (fjords with shallow tidewater glaciers), but that it would really help to make the discussion and conclusion more robust and more clear where the model has its strengths. Also the abstract should specify that this model is specific for fjords with shallow tidewater glaciers (because subglacial upwelling would change the effect of the meltwater considerably at deeper glacier fronts).

**Thank you for pointing that out. We added a clarification about the shallow grounding line depth in the Abstract, Discussion, and Conclusions and removed the generalizations.**

**Abstract (L13-16):**

**"Here, we present an analysis of satellite, meteorological, and SPM data as well as results of the coupled physical-biogeochemical model (1D GOTM-ECOSMO-E2E-Polar) with the newly implemented iSPM group, to show its impact on the ecosystem dynamics in the warming polar fjord (Hornsund, European Arctic) with the numerous shallow-grounded marine-terminating glaciers."**

**Discussion (L491-497):**

"Studies in deep Greenland fjords indicate that macronutrients were primarily supplied to the surface waters by mixing and not the transport from land with glacial meltwater as it was shown to have a relatively low nutrient load (Hopwood et al., 2020). However, Svalbard fjords are relatively shallow, and thus the upwelling pump might not be as efficient as for Greenland fjords or the shallower, nutrient-deficient waters might be transported (Hopwood et al., 2018). Furthermore, while macronutrient concentrations can be higher in the Arctic rivers, most of the discharge in Hornsund comes from marine-terminating glaciers (Błaszczyk et al., 2019)."

Conclusions (L515-518):

"Relatively well-studied areas adjacent to rapidly retreating marine-terminating glaciers in Hornsund are representative of similar coastal environments with shallow grounding line depth and, therefore, shed light on the formation and development of new marine habitats not only on a local but also on a regional scale."

Conclusions (removed sentence):

"Thus, the findings are potentially important for predictions in other regions such as Greenland, Patagonia, Alaska, and the Antarctic Peninsula, which experience temperatures close to, or above the melting point and hence are exposed to similar warming effects."